



# Assessment of the TROPOMI tropospheric NO₂ product based on airborne APEX observations

Frederik Tack[1], Alexis Merlaud[1], Marian-Daniel Iordache[2], Gaia Pinardi[1], Ermioni Dimitropoulou[1], Henk Eskes[3], Bart Bomans[2], Pepijn Veefkind[3], and Michel Van Roozendael[1]

[1]BIRA-IASB, Royal Belgian Institute for Space Aeronomy, Brussels, 1180, Belgium
[2]VITO, Flemish Institute for Technological Research, Mol, 2400, Belgium
[3]KNMI, Royal Netherlands Meteorological Institute, De Bilt, 3731, The Netherlands

*Correspondence to:* Frederik Tack (frederik.tack@aeronomie.be)

**Abstract.** Sentinel-5 Precursor (S-5P), launched in October 2017, carrying the TROPOspheric Monitoring Instrument (TROPOMI) nadir-viewing spectrometer, is the first mission of the Copernicus Programme dedicated to the monitoring of air quality, climate, and ozone. In the presented study, the TROPOMI tropospheric nitrogen dioxide (NO₂) L2 product (OFFL v1.03.01; 3.5 km x 7 km at nadir observations) has been validated over strongly polluted urban regions by comparison with coincident high-resolution Airborne Prism EXperiment (APEX) remote sensing observations (~75 m x 120 m). Satellite products can be optimally assessed based on (APEX) airborne remote sensing observations as a large amount of satellite pixels can be fully mapped at high accuracy and in a relatively short time interval, reducing the impact of spatio-temporal mismatches. In the framework of the S5PVAL-BE campaign, the APEX imaging spectrometer has been deployed during four mapping flights (26-29 June 2019) over the two largest urban regions in Belgium, i.e. Brussels and Antwerp, in order to map the horizontal distribution of tropospheric NO₂. For each flight, 10 to 20 TROPOMI pixels were fully covered by approximately 2800 to 4000 APEX measurements within each TROPOMI pixel. The TROPOMI and APEX NO₂ vertical column density (VCD) retrieval schemes are similar in concept. Overall for the ensemble of the four flights, the standard TROPOMI NO₂ VCD product is well correlated (R = 0.92) but biased negatively by -1.2 ± 1.2 x 10$^{15}$ molec cm$^{-2}$ or -14% ± 12%, on average, with respect to coincident APEX NO₂ retrievals. When replacing the coarse 1° x 1° TM5-MP a priori NO₂ profiles by NO₂ profile shapes from the CAMS regional CTM ensemble at 0.1° x 0.1°, the slope increases by 11% to 0.93, and the bias is reduced to -0.1 ± 1.0 x 10$^{15}$ molec cm$^{-2}$ or -1.0% ± 12%. When the absolute value of the difference is taken, the bias is 1.3 x 10$^{15}$ molec cm$^{-2}$ or 16%, and 0.7 x 10$^{15}$ molec cm$^{-2}$ or 9% on average, when comparing APEX NO₂ VCDs with TM5-MP-based and CAMS-based NO₂ VCDs, respectively. Both sets of retrievals are well within the accuracy requirement of a maximum bias of 25-50% for the TROPOMI tropospheric NO₂ product for all individual compared pixels. Additionally, the APEX data set allows the study of TROPOMI subpixel variability and impact of signal smoothing due to its finite satellite pixel size, typically coarser than fine-scale gradients in the urban NO₂ field. The amount of underestimation of peak plume values and overestimation of urban background values in the TROPOMI data is in the order of 1-2 x 10$^{15}$ molec cm$^{-2}$ on average, or 10% - 20%, in case of an urban scene.



# 1 Introduction

Sentinel-5 Precursor (S-5P), launched in October 2017, is the first of a series of atmospheric composition missions, planned within the European Commission's Copernicus Programme. It carries the TROPOspheric Monitoring Instrument (TROPOMI) nadir-viewing spectrometer as its single payload. TROPOMI provides measurements of the atmospheric composition with an unprecedented combination of accuracy, spatial coverage, and spatio-temporal resolution, introducing new opportunities such as studying the variability of pollutants at the scale of cities, in addition to the monitoring of the global distribution of gases.

The new sensor technology and retrieval approach requires carefully assessing the quality and validity of the generated data products to see if they meet their requirements in terms of accuracy and precision, by comparison with independent reference observations. The TROPOMI operational validation consists in routine quality control and long-term monitoring of the TROPOMI level-1 (L1) and level-2 (L2) products. This is performed within the European Space Agency (ESA) Mission Performance Center (MPC) in a semi-automatic way and based on a limited number of Fiducial Reference Measurements (FRM) available from ground-based reference networks, complemented by balloon and satellite observations. Large uncertainties however remain, mainly due to the mismatch in spatial representativeness of point-size stations and global satellite products. Routine validation is therefore complemented with campaign-based activities to provide a more in-depth, complete insight into the S-5P instrument performance and the fitness for purpose of its data products. A series of campaign activities have been identified in the S-5P Campaign Implementation Plan (S-5P CIP) (Tack et al., 2018), established to address key validation priorities.

On this basis, a S-5P validation campaign over Belgium (S5PVAL-BE), focusing on nitrogen dioxide (NO$_2$) column airborne observations, was identified as having high potential due to (1) the strong gradients in the NO$_2$ field over key Belgian cities, (2) the expertise built during the precursor BUMBA (Belgian urban NO$_2$ monitoring based on APEX remote sensing) campaigns over Belgium (Tack et al., 2017), and (3) the availability of the airborne prism experiment (APEX) hyperspectral imager and complementary ground-based infrastructure, such as mobile-DOAS, MAX-DOAS, and CIMEL stations. Aircraft remote sensing instruments, such as iDOAS (Heue et al., 2008), ACAM (Kowalewski and Janz, 2009), GeoTASO (Nowlan et al., 2016), AirMAP (Meier et al., 2017), Spectrolite (Vlemmix et al., 2017), SWING (Merlaud et al., 2018), GCAS (Nowlan et al., 2018) and APEX (Tack et al., 2017) are considered to be very valuable for satellite validation (van Geffen et al., 2018). The suitability of APEX to serve as independent reference for S-5P validation was assessed as part of the AROMAPEX project (Tack et al., 2019), a preparatory campaign activity focusing on the intercomparison of airborne atmospheric imaging systems (including APEX) and their mutual consistency, and the development of satellite validation strategies.

Tropospheric NO$_2$ is one of the principal trace gas products of TROPOMI. It is a key pollutant with a direct impact on human health and an important precursor of tropospheric ozone and particulate matter. NO$_2$ is primarily emitted as nitrogen monoxide (NO) and then rapidly oxidized to NO$_2$. In urbanized areas, the primary source is fuel combustion due to traffic, domestic heating and industrial activities. NO$_2$ is a short-lived species with a lifetime on the order of hours. Its distribution is characterised by a strong spatio-temporal variability when close to the emission sources. Due to its high spatial resolution,





TROPOMI is expected to be much more adequate to monitor short-scale urban NO$_2$ plumes than its predecessors, like GOME (Global Ozone Monitoring Experiment; 40 km x 320 km spatial resolution at nadir; 1995-2011; Burrows et al., 1999), SCIAMACHY (Scanning Imaging Absorption Chartography; 30 km x 60 km; 2002-2012; Bovensmann et al., 1999), OMI (Ozone Monitoring Instrument; 13 km x 24 km; 2004-present; Levelt et al., 2006), and GOME-2 (40 km x 80 km, 2007-present; Munro et al., 2016).

In this study, tropospheric NO$_2$ vertical column densities (VCDs), retrieved from high resolution APEX observations (~75 x 120 m$^2$), acquired during four flights (26-29 June 2019) over the two largest cities in Belgium, i.e. Brussels and Antwerp, have been compared with correlative retrievals from coincident S-5P overpasses. A single APEX flight typically covers a set of 10 to 20 TROPOMI pixels. The study focuses on the assessment of the TROPOMI L2 tropospheric NO$_2$ product (OFFL v1.03.01) in polluted regions, and more specifically on the accuracy and precision of the retrieved VCDs, and impact of intermediate products such as the slant column densities (SCDs), a priori NO$_2$ vertical profiles and surface reflectances (see Sect. 4 and 5). APEX provides a unique data set, allowing the study of TROPOMI subpixel variability, as well as the impact of signal smoothing (studied in Sect. 6) due to the finite satellite pixel size of TROPOMI, which is typically much larger than the fine-scale gradients in heterogeneous city plumes. The APEX spatial resolution is considerably higher than the typical resolution of spaceborne sensors. For example, one TROPOMI pixel (initially 3.5 km x 7 km at nadir observations and 3.5 km x 5.5 km since 6 August 2019) comprises approximately 2800 to 4000 APEX pixels.

Richter et al. (2014) discusses the challenges associated with the validation of tropospheric reactive gases. These challenges arise from the large spatio-temporal variability of short-lived reactive gases, the dependency of the products on different geophysical parameters (surface albedo, trace gases profiles, aerosols, etc.), different instrument sensitivities, and the presence of small signals close to the detection limit. In preparation of the Sentinel atmospheric missions S-5P and the forthcoming Sentinel-5 and Sentinel-4 missions (Ingmann et al., 2012), ESA has supported several projects to test newly developed airborne instruments and to develop satellite validation strategies, such as the AROMAT (airborne Romanian measurements of aerosols and trace gases; Meier et al., 2017; Merlaud et al., 2018; Merlaud et al., 2020) and AROMAPEX (Vlemmix et al., 2017; Tack et al., 2019) campaigns. The S5PVAL-BE campaign builds on the experience and lessons learned from these campaigns. For similar objectives, the National Aeronautics and Space Administration (NASA) has conducted a range of field campaigns including airborne imagers, such as the DISCOVER-AQ campaigns (https://discover-aq.larc.nasa.gov; Nowlan et al., 2016; Nowlan et al., 2018) and the KORUS-AQ campaign (https://wwwair.larc.nasa.gov/missions/korus-aq; Herman et al., 2018) in preparation of the geostationary TEMPO (Tropospheric Emissions: Monitoring Pollution; Zoogman et al., 2017), and the GEMS (Geostationary Environment Monitoring Spectrometer mission for Southeast Asia; Kim et al., 2017) missions, respectively.

This is the first publication assessing TROPOMI NO$_2$ retrievals over strongly polluted regions based on the comparison with airborne remote sensing observations. Earlier studies reporting on the validation of spaceborne observations based on airborne mapping data, such as Heue et al. (2005), Constantin et al. (2016) and Broccardo et al. (2018) have shown high



potential but are scarce, mainly due to the relatively coarse pixel footprint of TROPOMI's predecessors with respect to the area that can be covered with an airborne mapping spectrometer.

## 2 S5PVAL-BE campaign

Air pollution levels over Belgium are among the highest in Europe, with Brussels and Antwerp being key emission sources for anthropogenic nitrogen oxides ($NO_x = NO + NO_2$). In Antwerp, main $NO_x$ sources are related to (petro)chemical industry in the harbor area, while traffic emissions are dominant in Brussels. Strong gradients can be seen in Fig. 1 showing TROPOMI tropospheric $NO_2$ VCDs, ranging between 3 and 11 x $10^{15}$ molec $cm^{-2}$, observed over Belgium during a S-5P overpass on 27 June 2019 (orbit 8826). Markers indicate the five largest Belgian cities. Besides these $NO_2$ hotspots, long-range pollutant transport occurs regularly over Belgium. When wind is blowing from the north-northeast, plumes can be observed, emitted from the strongly industrialised Rhine-Ruhr valley in Germany and the port of Rotterdam in The Netherlands, which was the case on 27 June 2019.  Similarly, plumes can be observed, emitted from Lille and Dunkerque in France when the wind is south-southwest.

The S5PVAL-BE campaign took place in Belgium from 26 to 29 June 2019. In total four mapping flights, lasting between 1.5 and 2 hours each, took place on four consecutive days. The APEX hyperspectral imager was operated by the Flemish Institute for Technological Research (VITO) from a Cessna 208B Grand Caravan EX, with registration number HB-TEN, owned by Swiss Flight Services (SFS) at a nominal altitude of 6.5 km a.g.l. This is well above the planetary boundary layer (PBL), containing the bulk of the tropospheric $NO_2$. The aircraft followed a regular mapping pattern consisting of adjacent straight flight lines, with slightly overlapping footprints, alternately flown from south to north and from north to south, with the first flight line in the west. A sufficiently large area was covered over and around the city in order to capture the emission plumes downwind of the key sources and also to cover a large amount of TROPOMI pixels in order to have a statistically relevant data set. For each flight, approximately 10 to 20 TROPOMI pixels were covered for at least half their extension by APEX observations.

The coincident APEX mapping flights were scheduled to take place within one hour of the S-5P overpass, limiting the temporal variability between APEX and TROPOMI acquisitions to less than one hour. This requirement ensures largely (see section 5.2.2) that the same $NO_2$ field was observed by both the satellite and aircraft instrument. Flights took place in mostly cloud-free conditions and on days with good visibility. Only on 26 June (Flight #1), conditions were not fully optimal with few scattered clouds and some light haze and aerosols. Two flights took place over the city and harbor of Antwerp on 27 and 29 June and two flights over Brussels on 26 and 28 June. The flights covered variable meteorological and air quality conditions, as well as different overpass configurations, i.e. target area close to the TROPOMI nadir viewing direction (27 and 28 June 2019 with only one early-afternoon S-5P overpass) or closer to the edge of the swath (26 and 29 June 2019 with two early-afternoon S-5P overpasses). All relevant flight characteristics are provided in Table 1, as well as the meteorological and





environmental conditions during the flights. Note that the identifiers for the different flights (Flight #1 – Flight #4), as defined in Table 1, will be used in the continuation of this work to refer to the respective flights.

During all campaign days there was a light breeze between 2.6 and 3.7 m s$^{-1}$ at the surface, based on the average wind speed during the time of flight, and wind was usually blowing from the north-northeast, except for Flight #4 when there was a southeasterly wind. Wind and temperature data are collected from weather stations of the Royal Meteorological Institute of Belgium (RMI), i.e. Uccle station (50.8° N, 4.4° E, 100 m a.s.l.) for Brussels, and Stabroek station (51.3° N, 4.4° E, 4 m a.s.l.) for Antwerp, and measurements are averaged over the time of flight. Surface temperatures were high, ranging between 23° and 30° Celsius. All observations were performed close to solar noon and during the APEX acquisitions the solar zenith angle (SZA) ranged between 28° and 36° at maximum. The favourable high sun position during summer maximized the light backscattered to the sensor and minimized the signal smoothing occurring in case of shallow sun elevation angles (Lawrence et al., 2015). On the other hand, the overall NO$_2$ signal is generally slightly lower in summer time due to the shorter NO$_2$ lifetime.

Due to the local noon overpass of TROPOMI, we assume a deep and well-developed boundary layer and a good vertical dispersion of the anthropogenic emissions in the PBL due to turbulent mixing from surface heating. During the overpasses, a PBL height between 700 and 900 m was retrieved from the backscatter profiles of a Vaisala CL51 ALC ceilometer operated by RMI in Uccle. A low aerosol optical thickness (AOT level 1.5) of less than 0.15, at 500 nm, was measured during Flight #2 – Flight #4 by a CIMEL sun photometer at the AERONET station (Holben et al., 1998) in Uccle. During Flight #1, an AOT of 0.51 at 500 nm was observed. On average the retrieved AOT was 0.17 for June 2019. Note that the Uccle station is located south of Brussels, so for 26 to 28 June we assume that the site was downwind of the Brussels city center and thus in a semi-polluted area. The CIMEL observations are largely consistent with measurements performed with a handheld Model 540 Microtops II sun photometer from Solar Lights (Porter et al., 2001). Measurements were performed from a car, looping around the city during the APEX overpasses. An average AOT (440 nm) of 0.65, 0.19 and 0.16 was observed by the Microtops on 26, 27 and 28 June, respectively.

## 3 Observation systems

### 3.1 S-5P and the TROPOMI payload

The TROPOMI instrument is a nadir-viewing pushbroom imaging spectrometer, and was built by a joint venture between the Netherlands Space Office (NSO), Royal Netherlands Meteorological Institute (KNMI), Netherlands Institute for Space Research (SRON), Netherlands Organisation for Applied Scientific Research (TNO), Airbus Defence and Space Netherlands, and ESA. TROPOMI builds upon a rich heritage from similar instruments, such as SCIAMACHY (Bovensmann et al., 1999) on ESA's ENVISAT and OMI (Levelt et al., 2006) on NASA's AURA satellite. The main objective of the S-5P mission is to perform atmospheric measurements, relating to air quality, climate forcing, ozone and UV radiation. S-5P bridges the gap in



continuity of observations between its ESA predecessors (GOME and SCIAMACHY) and the forthcoming Sentinel-5 and Sentinel-4 missions, planned to be launched in 2023.

The TROPOMI instrument consists of four spectrometers covering the UV-VIS-NIR-SWIR wavelength ranges at a spectral resolution of 0.45-0.65 nm in the UV-VIS range. S-5P is in a near-polar, sun-synchronous orbit of 824 km in altitude with an ascending node equatorial crossing at 13:30 Mean Local Solar time. The entrance telescope allows for a wide field of view (FOV) of 108°, corresponding to a swath width of approximately 2600 km, providing daily global coverage with a ground pixel size of approximately 3.5 km x 7 km at nadir (3.5 km x 5.5 km since 6 August 2019). For a full technical description, we refer to Veefkind et al. (2012), Loots et al. (2017) and Kleipool et al. (2018). The TROPOMI key specifications are provided and compared with the APEX specifications in Table 2.

### 3.2 APEX airborne imager

The APEX instrument is a pushbroom imaging spectrometer, designed and developed on behalf of ESA by a Swiss-Belgian consortium (Itten et al., 2008; D'Odorico, 2012; Schaepman et al., 2015). Currently, APEX is jointly owned and operated by the Remote Sensing department of the Flemish Institute for Technological Research (VITO-TAP, Mol, Belgium) and the Remote Sensing Laboratories from University of Zurich (RSL-UZH, Zurich, Switzerland). APEX records backscattered solar radiation in the visible, (near-)infrared regions of the electromagnetic spectrum, covering the 370 to 2540 nm wavelength range in two channels. In this study, only data from the VNIR channel (370-970 nm) was used. The radiance is spectrally dispersed by a prism. Hence, the full width at half maximum (FWHM) is a strongly non-linear function of the wavelength, broadening from 1.5 to 3 nm FWHM in the visible spectral range. The CCD (charge-coupled device) 14 bit depth area detector records data in 1000 pixels across-track (spatial dimension) and 335 bands in the spectral dimension. Based on the across-track field of view (FOV) of 28°, a swath width of 3.2 km is obtained at a nominal flight altitude of 6.5 km a.g.l. The native spatial resolution of 3 m x 4 m, across- and along-track respectively, is spatially aggregated to a resolution of approximately 75 m x 120 m in order to increase the signal to-noise ratio (SNR), while retaining sufficient spatial detail for atmospheric composition measurements (Tack et al., 2017). The APEX optical unit is enclosed by a thermoregulated box, while the pressure in the spectrometer is kept at 200 hPa above ambient pressure.

In Table 2, the provided $NO_2$ SCD detection limits are approximated by the average 1-sigma slant error on the DOAS fit, as instrument noise is the dominant source of errors in the spectral fitting. Using the same definition, $NO_2$ SCD detection limits are estimated to be 5.6 x $10^{14}$ molec cm$^{-2}$ for TROPOMI retrievals and 2.6 x $10^{15}$ molec cm$^{-2}$ for APEX retrievals at its native resolution of 75 m x 120 m (See Sect. 5.2.1). However, in Sect. 5.2.2 spatio-temporal coinciding TROPOMI and APEX $NO_2$ VCD grids are quantitatively compared by spatial averaging of all APEX $NO_2$ VCDs within each TROPOMI pixel footprint. One nadir TROPOMI pixel corresponds to approximately 2700 APEX pixels, providing good statistics in the comparison. Spatial aggregation of APEX retrievals results in a decrease of its random uncertainty. Following Poisson statistics and assuming only photon noise, the noise is expected to decrease with the square root of the number of aggregated retrievals,





resulting in a noise reduction by a factor 52 or a noise level of 5.0 x $10^{13}$ molec cm$^{-2}$ on the aggregated APEX pixels. This is approximately one tenth of the TROPOMI random error. The effective APEX noise level is, however, expected to be slightly larger as the noise reduction due to spatial binning does not completely follow shot noise statistics due to occurring dark current and read-out noise and systematic errors in the DOAS fit.

Several studies have demonstrated the capabilities of APEX for atmospheric trace gas retrieval applications, in particular high-resolution mapping of the $NO_2$ variability over polluted regions (Popp et al., 2012; Kuhlmann et al., 2016; Tack et al., 2017; Tack et al., 2019).

## 4 NO₂ VCD retrieval algorithm

### 4.1 TROPOMI NO₂ processor

The TROPOMI $NO_2$ processor is based on the DOMINO v2 (Dutch OMI $NO_2$ data products of KNMI for OMI; Boersma et al., 2011) and QA4ECV (Quality Assurance for Essential Climate Variables; Boersma et al., 2018) processing systems, with a number of differences related to specific TROPOMI characteristics. The processor is based on a retrieval-data assimilation-modelling system using the 3-D global TM5-MP chemistry transport model (CTM) (Williams et al., 2017) and is based on a 3-step approach:

(1)  The retrieval of $NO_2$ slant columns, being the $NO_2$ concentration integrated along the effective light path, by application of the Differential Optical Absorption Spectroscopy (DOAS) baseline method (Platt and Stutz, 2008) on the Level-1b radiance and irradiance TROPOMI spectra. The DOAS retrieval follows a non-linear fitting approach similar to the one used for OMI (Boersma et al., 2011; van Geffen et al., 2015). Key retrieval parameters are provided in Table 3. Resulting SCDs are dependent on the optical light path through the atmosphere and thus on the viewing
geometry, the assumed state of the atmosphere and solar radiative transfer.

(2)  Separation of the total slant column into its tropospheric and stratospheric contributions, based on data assimilation of the SCDs in the TM5-MP CTM (Williams et al., 2017).

(3)  Conversion of the retrieved SCDs into VCDs by application of appropriate air mass factors (AMFs). AMFs express the relationship between SCDs and VCDs, accounting for the effects of the viewing and sun geometry, $NO_2$ vertical
distribution, surface albedo, cloud fraction, cloud height, aerosol scattering, and terrain height. They are obtained by the integrated product of (1) altitude-dependent AMFs (or box AMFs) expressing the vertical sensitivity of the measurement, and (2) daily $NO_2$ vertical profiles from the TM5-MP model on a $1° \times 1°$ grid and covering 34 vertical layers. Box AMFs are computed based on the Doubling-Adding KNMI (DAK version 3.2) radiative transfer model (RTM) (De Haan et al., 1987; Stammes et al., 2001). TROPOMI surface albedo is based on a climatology made from
5 years of OMI data, aggregated to a grid of $0.5° \times 0.5°$ (Kleipool et al., 2008). For $NO_2$ retrievals, the lambert equivalent reflectance (LER) at 440 nm is used. The LER is defined as the required reflectance of an isotropic surface





needed to match the observed top of the atmosphere (TOA) reflectance in a pure Rayleigh scattering atmosphere under cloud free conditions and no aerosols. Cloud parameters are retrieved based on the fast retrieval scheme for clouds from the oxygen A band algorithm (FRESCO+; Wang et al., 2008).

For a full description of the TROPOMI NO$_2$ retrieval algorithm, we refer to the algorithm theoretical basis document (ATBD) of the total and tropospheric NO$_2$ data products (van Geffen et al., 2018) and the recent study of van Geffen et al. (2020). Note that in the continuation of this work, NO$_2$ VCD$_{TROPO}$ refers to the TROPOMI tropospheric NO$_2$ VCD product based on the standard TM5-MP profiles.

## 4.2 APEX NO$_2$ processor

The APEX NO$_2$ VCD retrieval scheme is similar in concept to the TROPOMI one and is discussed in detail in Tack et al.
(2017; 2019). The DOAS spectral fit is based on the QDOAS software (Fayt et al., 2016) applied in the 470-510 nm spectral range, optimal for NO$_2$ retrieval from APEX. Note that interference with unidentified instrumental artefacts or features prevents us from extending the fitting window to wavelengths lower than 470 nm as discussed in Popp et al. (2012) and Tack et al. (2017). Key parameters for the NO$_2$ SCD retrieval are provided in Table 3. NO$_2$ box AMFs have been calculated with the LIDORT 2.6 RTM (Spurr, 2008). Sun and viewing geometry, defined by the SZA, viewing zenith angle (VZA), and relative
azimuth angle (RAA) are computed by the APEX ortho-rectification module (Vreys et. al, 2016) for each observation. Pressure and temperature atmospheric profiles are taken from the AFGL standard atmosphere for mid-latitude summer (Anderson et al., 1986). Aerosol extinction profiles (AEPs) were constructed from the AOT and PBL height observations, measured by the CIMEL and ceilometer, respectively, during the respective flights (see Table 1). As APEX is radiometrically calibrated, a surface reflectance product can be retrieved from the at-sensor radiances by application of an atmospheric correction algorithm
(Sterckx et al., 2016). Total AMFs are computed from the box AMFs based on integration along an a priori NO$_2$ box profile, with constant mixing ratio in the PBL and taking the PBL height from the ceilometer observations (see Table 1). In the continuation of this work, NO$_2$ VCD$_{APEX}$ refers to the retrieved APEX tropospheric NO$_2$ VCD product.

## 4.3 AMF dependence on key RTM parameters

### 4.3.1 A priori NO$_2$ profile

A priori NO$_2$ profile shapes used in the TROPOMI retrieval algorithms are specified using the TM5-MP CTM, which is an improved version of the TM4 CTM operated for the OMI DOMINO v2.0 product. TM5-MP has a finer spatial resolution (1° x 1°), updated information on NOx emissions, and an improved description of relevant physical (photolysis rate constants) and chemical (reaction rate constants) processes (van Geffen et al., 2018). However, highly polluted areas typically exhibit strong NO$_2$ vertical and horizontal gradients (see e.g. Dieudonné et al., 2013; Ialongo et al., 2019; Zhao et al., 2019; Dimitropoulou





et al., 2020; Pinardi et al., 2020). The sharp gradients between pollution plumes and background areas cannot be resolved properly at the horizontal scale of the model (~100 km x 100 km). In Dimitropoulou et al. (2020), TROPOMI tropospheric $NO_2$ VCDs were recalculated based on high-resolution MAX-DOAS profiles, while in Ialongo et al. (2019) a priori $NO_2$ profiles were extracted from the Copernicus atmospheric monitoring service (CAMS) regional CTM (Marécal et al., 2015;

https://www.regional.atmosphere.copernicus.eu). These transformations generally led to increased $NO_2$ VCDs, resulting in a better agreement with reference ground-based measurements. In this study, a custom TROPOMI tropospheric $NO_2$ product was also prescribed, based on $NO_2$ profile shapes from the CAMS-regional CTM ensemble. CAMS $NO_2$ profiles, being a merge of CAMS-regional (0.1° x 0.1°; surface to 3 km altitude in seven layers; hourly data) and CAMS-global (0.4° x 0.4°; 3 km to TOA; 3-hourly data), analysed at the 0.1° grid of CAMS-regional, were used to recompute the tropospheric AMFs and

corresponding TROPOMI $NO_2$ VCDs, referred to as $VCD_{TROPO-CRE}$ in the continuation of this work. In general we find that the VCDs are increased by about 5% to 40% over the Brussels-Antwerp regions, depending on the day and location (see Fig. 2). In the absence of $NO_2$ hotspots and plumes, the impact of changing the a priori profile is small. Both the standard and the custom TROPOMI $NO_2$ product are compared with airborne APEX mapping data in Sect. 5.2.

    For the APEX retrievals, AEPs and a priori $NO_2$ profiles were constructed from the AOT and PBL height observations,

as discussed in Sect. 4.2. In order to yield retrievals independent from the satellite, box profiles were used instead of the TROPOMI TM5-MP profiles, as displayed in Fig. 3a. When TM5-MP profiles would be applied as a priori for the APEX retrievals, the AMF would increase with 9% on average, which is consistent with a similar sensitivity study reported in Tack et al. (2017). For the APEX retrievals, we assumed a well-mixed $NO_2$ and aerosol box profile scenario and urban aerosols with a high single-scattering albedo (SSA) of 0.93. This causes a multiple scattering scenario and an enhancement of the optical

path length in the $NO_2$ layer, and results in an increase in the AMF. When instead considering a no aerosol scenario for the APEX retrievals, the AMF drops by 10% on average. We assume that the opposing effects of using (1) a priori profile shape assumptions different from the TROPOMI retrievals and (2) different aerosol assumptions tend to cancel each other out in the APEX retrievals.

    Box AMFs were computed and plotted in Fig. 3b for APEX, operating at 6.5 km a.g.l., and TROPOMI, for both a low

and high surface reflectance scenario and with fixed values for the other RTM parameters. The box AMFs describe the sensitivity of the observations as a function of altitude (Wagner et al., 2007). The shapes of both TROPOMI and APEX box AMFs are similar below the aircraft altitude (6.5 km a.g.l.), but APEX has a higher sensitivity. As can be seen, the nadir-looking airborne instrument has a peak in sensitivity in the layer directly under the sensor. Above the airborne platform, the sensitivity to $NO_2$ converges rapidly with increasing altitude to a constant box AMF of 1.6, a value which corresponds to the

geometrical AMF at the SZA of 50° assumed for this simulation. Due to scattering and absorption, the sensitivity decreases towards the ground surface where the bulk of the tropospheric $NO_2$ is residing. The decrease in sensitivity is stronger for TROPOMI, due to the larger probability of scattering above the $NO_2$ layer. For a low albedo case, i.e. 0.02, the surface box AMF is about 50% larger in case of APEX, while this is ~15% for a very high albedo case, i.e. 0.2. For the ensemble of the





four data sets, the total AMF is 1.1 ± 0.1 on average for TROPOMI retrievals and 1.7 ± 0.1 for APEX retrievals, or approximately 50% higher.

### 4.3.2 Surface reflectance

The surface albedo used in the TROPOMI retrievals is currently based on a climatology made from 5 years of OMI data,
aggregated to a grid of $0.5° \times 0.5°$, as discussed in Sect. 4.1. In this section, we first compare the TROPOMI albedo at 440 nm with the surface reflectance product retrieved from the APEX at-sensor radiance at 490 nm (see Sect. 4.2). Similar to the assessment of coincident $NO_2$ VCDs in Sect. 5.2.2, the spatio-temporal coinciding TROPOMI and APEX albedo grids are quantitatively compared by spatial averaging of all APEX albedo values within each TROPOMI pixel footprint for the ensemble of the APEX data sets. The latter is defined by the pixel corner coordinates provided in the L2 product, while the
APEX albedo locations are defined by their respective pixel center coordinates. TROPOMI pixels currently take the albedo values from the coarse OMI LER (~50 km x 50 km), implying that groups of neighboring TROPOMI pixels are assigned the same value. As a result, usually one APEX data set over a particular city covers only one to two different OMI LER albedo values. As APEX measures the albedo at high resolution (75 m x 120 m), we consider it as a good approximation of the effective albedo. By comparing the APEX albedo to the TROPOMI albedo, we can have an indication of the effective albedo
variability over an urban area and how this is smoothed out in the TROPOMI/OMI LER, due to its coarser resolution.

      Analysing the ensemble of the four acquired APEX data sets provided in Table 1, the APEX albedo is 0.040 on average and the variability within one TROPOMI pixel, expressed as the standard deviation (SD), is 0.022 on average or ~55%, but can be up to 100% for certain pixels (see Fig. 4). When considering the entire APEX scenes instead of single pixels, the variability of the APEX-derived albedo, resampled at TROPOMI resolution, is 0.012 on average or ~30%, with values ranging
between 0.015 and 0.065. The OMI-based TROPOMI albedo variability is low, i.e. 0.001, as only 4 different 0.5° OMI LER pixels are sampled over the APEX scenes. The strong effective albedo variability over urban areas, as illustrated by the APEX albedo, is not captured by the OMI LER. This is likely to introduce a noise in the $NO_2$ VCD retrieval since this variability is not accounted for in the computed AMFs. In Sect. 5.2.2 it is shown that the comparison of APEX with coincident TROPOMI tropospheric $NO_2$ SCDs exhibits a slightly smaller spread than when comparing APEX and TROPOMI VCDs.

The albedo for coincident TROPOMI pixels over the APEX scenes is 0.051, on average, or 0.011 (~27%) higher than APEX. This is somewhat surprising at first glance as one would expect that high albedo values, typically observed over urban areas (Heiden et al., 2007), would be smoothed out in the OMI LER low-resolution albedo product and that this would result in a lower overall albedo when compared to the high-resolution APEX product. However, Kleipool et al. (2008) discusses that a statistical analysis approach is used to yield a climatologically averaged reflectance in the OMI LER, instead of using an
absolute minimum reflectance method or so-called minimum Lambertian equivalent reflectance (MLER). The statistical analysis approach results in a higher reflectance value than provided by the MLER. This is to take into account the presence of boundary layer haze and persistent cloud features. It seems that for clear-sky conditions, the OMI LER overestimates the



surface reflectance and that for these conditions the MLER would be a better approximation. Over Belgium, OMI MLER (not provided in the TROPOMI L2 $NO_2$ product) is approximately 0.005 lower than the OMI LER reflectance value, which would reduce the overestimation of TROPOMI reflectance to 0.006 when compared to APEX for the clear-sky flights. According to Boersma et al. (2004), for albedo values smaller than 0.200, an overestimation of the albedo by 0.005-0.010 can result in a 5-

10% increase of the tropospheric AMF, and thus in a potential underestimation of the retrieved TROPOMI $NO_2$ VCD.

The APEX and TROPOMI albedo have been both compared with Moderate Resolution Imaging Spectroradiometer (MODIS) albedo data, and more specifically with the MODIS MCD43A3 black-sky albedo daily L3 500 m v006 product at 470 nm. Coincident APEX and MODIS albedo pixels are compared for the data set acquired over Antwerp on 27 June 2019 and the scatter plot is shown in Fig. 5a. The regression analysis shows a high correlation (R = 0.96) and a slope close to unity

on a total of 2800 compared pixels, while the absolute difference is smaller than 0.005, on average. When comparing TROPOMI and MODIS albedo, both data sets are regridded to 0.5°, being the gridsize of the OMI LER. Albedo pixels are compared for the whole of Belgium on 27 June 2019 and the scatter plot is shown in Fig. 5b. The dynamic range is much lower than for the comparison between APEX and MODIS albedo and high albedo values (> 0.06), typically observed over urban areas, are smoothed out. The regression analysis shows a lower correlation (R = 0.84) and the TROPOMI albedo is

approximately 0.012 higher than MODIS. Similar statistics were found when comparing the data sets acquired on the other campaign days.

The observed overestimation of the OMI LER seems to be consistent with comparison studies reported in Kleipool et al. (2008). In the study, the OMI LER has been assessed for the entire globe by comparison with a similar LER map, based on data from the Total Ozone Mapping Spectrometer (TOMS) at 331, 340, 360, and 380 nm. The TOMS LER was

approximately 0.015 lower than the OMI LER on average. GOME albedo at 335, 380, 440, and 494 nm was ~0.005 lower on average. The OMI LER was approximately 0.020 higher than the black-sky albedo, derived from MODIS at 470 nm. According to Kleipool et al. (2008) this is partly related to viewing geometry effects of the bidirectional reflectance distribution function (BRDF) of the surface. The TROPOMI and MODIS reflectance products are also not provided at the same wavelength and a statistical analysis approach is used to determine the reflectance value, instead of the OMI MLER.

Even if a direct comparison of different albedo products is not trivial due to BRDF-effects and albedo wavelength dependencies, among others, there is an indication that the OMI LER is overestimating the effective albedo in certain conditions, requiring a revision of the product and algorithm. Retrievals over strongly polluted areas also require an albedo product at higher resolution in order to resolve the typically strong albedo variability. A global gapless geometry-dependent LER (G3_LER) daily map product at 0.1°, retrieved from the TROPOMI L1B radiances, is currently under development and

discussed in Loyola et al. (2020). Also KNMI is working on a new TROPOMI LER product, extended compared to the OMI LER by including a viewing angle dependency, and will become available after the L1B product has been reprocessed to v2. As soon as a TROPOMI LER product becomes available, and its impact has been tested, this will be implemented to replace the OMI albedo climatology (Henk Eskes, personal communication, 15 March, 2020).



Furthermore, for the NO$_2$ retrieval a surface albedo adjustment scheme has already been implemented and will become operational from v2.0 onwards (upgrade planned for the second half of 2020; Eskes et al., 2020). In this approach the reflectivity measured in the NO$_2$ fitting window will be compared with a computed reflectivity based on the LER climatology. In case the observed reflectivity is lower, the albedo value will be reduced to match the observation, and the AMF will be computed with the adjusted albedo. This approach should remedy part of the shortcomings of the current albedo climatology.

### 4.3.3 Cloud fraction

Due to the cloud-free conditions for Flight #2 to #4, cloud parameters do not contribute to the uncertainties here. Nevertheless, the effective cloud radiance fraction for the NO$_2$ retrieval window at 440 nm, computed by FRESCO (Wang et al., 2008) and provided in the L2 product, was checked. For Flights #2 to #4, a cloud cover of less than 0.5% on average was computed over Belgium. During Flight #1, scattered clouds were present and a cloud fraction of on average 12% was computed over Belgium.

A small cloud fraction of 12% indicates that there is more scattering in the atmosphere than computed based on the LER value. In the TROPOMI NO$_2$ retrieval such small "cloud fractions" are used to implicitly compensate for aspects like too small LER values (e.g. often the case over cities which have a higher reflectivity than the surroundings not resolved in the OMI map), or the presence of scattering aerosols, haze or residual clouds. Ideally the cloud pressure will indicate the altitude at which the scattering takes place. In practice this is a challenge because cloud pressure uncertainties are large for small cloud fractions.

## 5 Results

### 5.1 Analysis of the APEX NO$_2$ VCD grid product

The retrieved APEX NO$_2$ VCD maps are provided in Figs. 6 and 7 for the Antwerp (Flight #2 and #4), and Brussels (Flight #1 and #3) campaign, respectively. Flight characteristics, and meteorological and environmental parameters of the four APEX flights were already discussed in Sect. 2 and are summarized in Table 1. They assist the geophysical interpretation of the observed NO$_2$ field. On the maps, white dots indicate the key point sources which are mostly chimney stacks from the prevailing petrochemical industry in the harbour of Antwerp. They are retrieved from the emission inventory 2017, provided by the Belgian Interregional Environmental Agency and a threshold was set at a minimum emission of 10 kg of NOx per hour in order to discriminate and visualize the main emitters. Key line sources such as the highways and city ring roads are indicated by white lines. TROPOMI tropospheric NO$_2$ VCD retrievals are overlayed as color-coded polygons, defined by the pixel corner coordinates provided in the L2 product, and exhibit in general a good consistency with the APEX retrievals. However, elevated levels of NO$_2$ from isolated hotspots or narrow and confined plumes, visible in the APEX retrievals, cannot be spatially resolved anymore by TROPOMI and are averaged out within the TROPOMI pixel. This is for example the case for the plume





in the north of the APEX data set acquired over Antwerp on 27 June 2019. This smoothing effect will be studied in more detail in Sect. 6.

The spatial resolution of the APEX retrievals allows to reveal the urban fine-scale $NO_2$ horizontal variability, and to resolve individual emission sources. Strong patterns of enhanced $NO_2$ can be discerned and linked to the key point and line

sources. The maps reveal that the $NO_2$ field is highly variable in urban areas in both space and time. The $NO_2$ VCDs retrieved by APEX range between 1 and 40 x $10^{15}$ molec $cm^{-2}$ in Antwerp with an average of 7.6 $\pm$ 3.8 x $10^{15}$ molec $cm^{-2}$ for Flight #2 and 9.9 $\pm$ 6.1 x $10^{15}$ molec $cm^{-2}$ for Flight #4. In Antwerp, the anthropogenic emissions are mainly related to industrial activities in the harbour. Some fine-scale plumes from individual stacks can be observed, while clusters of stacks contribute together to larger plumes. The observed plumes, narrow and confined close to their sources, are transported downwind for several tens of

kilometers, as can also be observed in the TROPOMI retrievals (see Fig. 1). The primary emitted pollutant is NO, which is typically oxidised to $NO_2$ after entering the atmosphere. Further downwind, the $NO_2$ mixes and accumulates in the PBL and the plumes get more dispersed. Part of the emissions can also be related to traffic: increased values can be observed in the city center of Antwerp as well as along and downwind from the ring road R1 and junctions with the key highways E313 in the east and E19 in the west. Note that the main emission sources are largely the same as observed during previous APEX flights over

Antwerp, as discussed in Tack et al. (2017).

Although June 29 is a Saturday, the $NO_2$ VCDs observed over the Antwerp harbour are slightly higher than on June 27, both in the APEX and TROPOMI data. However, when averaging the $NO_2$ levels for the whole of Belgium, TROPOMI observes a slightly lower tropospheric $NO_2$ VCD on June 29 (3.3 $\pm$ 1.2 x $10^{15}$ molec $cm^{-2}$) than on June 27 (3.8 $\pm$ 1.3 x $10^{15}$ molec $cm^{-2}$). Note that some instrumental problems were encountered during the flight on June 29. The APEX instrument

switched to an unstable state during the acquisition of the first three flight lines in the East over the city center of Antwerp, hampering the application of the retrieval algorithm on the corrupted spectra. The problem occurred as well in some parts of flight lines 4 to 6 explaining the gaps in the data set. The reasons for these instrument instabilities are currently unidentified.

The observed $NO_2$ VCDs range between 1 and 24 x $10^{15}$ molec $cm^{-2}$ in Brussels with an average of 9.8 $\pm$ 4.2 x $10^{15}$ molec $cm^{-2}$ for Flight #1 and 6.9 $\pm$ 2.8 x $10^{15}$ molec $cm^{-2}$ for Flight #3. Here, the observed anthropogenic emissions are

predominantly related to traffic and relatively small-scale industrial activity along the Brussels canal, indicated by the blue line. In this area, a considerable contribution is expected to come from a waste-to-energy plant. The station is indicated by the white dot in the north of the data set and is emitting approximately 15 kg of NOx per hour according to the emission inventory (2017). For Flight #1, a plume originating from the Antwerp harbor and transported over the eastern part of Brussels can be observed in both the TROPOMI and APEX data. A large city plume, moving downwind in southwestern direction, can be

observed in the Flight #3 data, as well as hotspots near the Brussels city center and increased $NO_2$ levels along the R0 Brussels ring road and the junctions with the key highways E40 and E19. The R0 is one of the busiest highways in Belgium with traffic volumes of more than 70 000 cars per day. $NO_2$ hotspots can also be observed in the area of the Brussels international airport in the northeast, related to aircraft and airport traffic operations.



## 5.2 Assessment of the TROPOMI NO₂ product

### 5.2.1 Error budget - precision assessment

The TROPOMI L2 tropospheric NO₂ product (OFFL v1.03.01) has been assessed based on independent high-resolution airborne APEX data, acquired over the target areas within one hour of the S-5P overpass time. The accuracy and precision requirements for the TROPOMI L2 products have been formulated by the L2 Quality Working Group (QWG) and agreed on with the S-5P Mission Advisory Group (MAG). The accuracy of the tropospheric NO₂ VCD product is targeted to be around 25-50%, with a precision of 0.7 x $10^{15}$ molec cm⁻² (Fehr, 2016).

The TROPOMI NO₂ processing chain allows to provide a realistic error budget. The total TROPOMI tropospheric NO₂ VCD error, σVCD$_{TROPO}$, is driven by (1) error propagation of the slant column errors, (2) errors associated with the separation of the stratospheric and tropospheric contributions, and (3) tropospheric AMF errors (van Geffen et al., 2018). The overall error in the TROPOMI tropospheric NO₂ columns can be quantified based on Boersma et al. (2004):

$$\sigma_{VCD_{TROPOi}} = \sqrt{\left(\frac{\sigma_{SCD_{TROPOi}}}{AMF_{TROPOi}}\right)^2 + \left(\frac{\sigma_{SCD_{TROPOi}^{strato}}}{AMF_{TROPOi}}\right)^2 + \left(\frac{SCD_{TROPOi} - SCD_{TROPOi}^{strato}}{AMF_{TROPOi}^{2}} \times \sigma_{AMF_{TROPOi}}\right)^2} \qquad (1)$$

The overall error variance is provided for each retrieval in the L2 product and is fully described in van Geffen et al. (2018). Analysis of the TROPOMI L2 NO₂ VCDs, coinciding with the APEX data sets, reveals a mean VCD and absolute error of 6.8 (VCD$_{TROPO}$) and 7.9 (VCD$_{TROPO-CRE}$) ± 2.1 x $10^{15}$ molec cm⁻² or a relative error of approximately 31% and 27%, respectively. The maximum relative error observed was 42%. In Fig. 8, the vertical error bars indicate the overall error for each TROPOMI VCD retrieval. In general, larger relative errors are seen mostly over semi-background areas, reflecting mainly uncertainties in the slant and stratospheric column retrieval. Over polluted regions, the absolute errors increase while the relative errors drop. Here, the retrievals are largely dominated by systematic errors in the computation of the AMFs. These are related to uncertainties in the assumptions made for the RTM parameters with respect to the true atmospheric state, and are dominated by the NO₂ profile shape, surface albedo and cloud parameters (cloud fraction and height). Uncertainties propagated due to the NO₂ profile assumptions and surface albedo have been discussed in section 4.3. The effect of clouds, however, was not considered in this study as data acquisition took place in mostly clear-sky conditions.

The TROPOMI precision is targeted to be better than 7.0 x $10^{14}$ molec cm⁻² (Fehr, 2016). We looked into the fitting error, σSCD$_{TROPOi}$, as a proxy to assess compliance with the precision requirement, as instrument noise is the dominant source





of errors in the spectral fitting of TROPOMI Level-1b spectra. Averaged over the four campaign days over Belgium, the precision is estimated to be $5.6 \pm 0.4 \times 10^{14}$ molec cm$^{-2}$, thus well within the requirement. This is consistent with an assessment performed over the Pacific Ocean and discussed in van Geffen et al. (2018), reporting precision levels between 5.0 and 6.0 x $10^{14}$ molec cm$^{-2}$. Note that the TROPOMI noise level is approximately 30% lower than the initial OMI noise level (as measured

in 2005). This is due to the higher radiometric SNR of TROPOMI, which is around 1400−1500 (~900-1000 for OMI) for an individual Level-1b spectrum in the 400−500 nm range (van Geffen et al., 2018).

Like for TROPOMI, the overall error on the retrieved APEX NO$_2$ VCDs, $\sigma$VCD$_{APEX}$, is dominated by uncertainties related to the DOAS fit and AMF computation. Due to the negligible spatio-temporal variability of the stratospheric NO$_2$ field in the time between the acquisition of the reference spectrum and the measurements, i.e. less than one hour, the stratospheric

NO$_2$ contribution to the signal is expected to cancel out in case of APEX retrievals and is consequently not treated as a key error source. On the other hand, the error originating from the estimation of the NO$_2$ residual amount in the reference spectrum, $\sigma$SCD$_{ref}$, can be considerable as discussed in Tack et al. (2017). SCD$_{ref}$ is derived from co-located mobile-DOAS measurements and the error can be up to $1.8 \times 10^{15}$ molec cm$^{-2}$. The overall error in the APEX tropospheric NO$_2$ columns can be quantified based on the following error propagation method and is discussed in detail in Tack et al. (2017):

$$\sigma_{VCD_{APEXi}} = \sqrt{\left(\frac{\sigma_{DSCD_{APEXi}}}{AMF_{APEXi}}\right)^2 + \left(\frac{\sigma_{SCD_{ref}}}{AMF_{APEXi}}\right)^2 + \left(\frac{SCD_{APEXi}}{AMF_{APEXi}{}^2} \times \sigma_{AMF_{APEXi}}\right)^2} \qquad (2)$$

Analysis of all coincident APEX NO$_2$ VCD reference measurements for the ensemble of the four flights reveals a mean VCD and absolute error of $8.0 \pm 2.3 \times 10^{15}$ molec cm$^{-2}$ or a relative error of approximately 29%. This is consistent with the retrieval errors found during previous APEX campaigns, e.g. BUMBA (Tack et al., 2017) and AROMAPEX (Tack et al.,

2019) and is also in line with the typical error found for similar airborne hyperspectral imaging instruments (Tack et al., 2019). Spatio-temporal coinciding TROPOMI and APEX NO$_2$ VCD grids are quantitatively compared in Sect. 5.2.2 by spatial averaging of all APEX NO$_2$ VCDs within each TROPOMI pixel footprint, resulting in a decrease of the overall random uncertainty on APEX retrievals. As discussed in Sect. 3.2, the average APEX noise level is expected to decrease from ~2.6 x $10^{15}$ to ~5.0 x $10^{13}$ molec cm$^{-2}$ after spatial aggregation. Propagating this into the mean APEX VCD error, $\sigma$VCD$_{APEX}$, the latter

is expected to be reduced from 2.3 to $1.6 \times 10^{15}$ molec cm$^{-2}$, or a reduction of the relative error from ~29% to ~21% on the retrieved APEX NO$_2$ VCDs. The noise reduction has the biggest impact on retrievals over (urban) background areas, as the errors here are dominated by uncertainties in the slant column retrieval.





### 5.2.2 Comparison of coincident NO$_2$ VCDs – accuracy assessment

Satellite products can be optimally assessed based on airborne observations as a large amount of satellite pixels can be fully mapped at high resolution in a relatively short time interval, reducing the impact of spatio-temporal mismatches. The spatio-temporal coinciding TROPOMI and APEX NO$_2$ VCD grids are quantitatively compared by spatial averaging of all APEX NO$_2$

VCDs within each TROPOMI pixel footprint. The latter is defined by the pixel corner coordinates provided in the L2 product, while the APEX VCD locations are defined by their respective pixel centre coordinates. Note that TROPOMI pixels are only considered in the further analysis when they are covered for more than 50% by APEX pixels in order to reduce undersampling. Prior to the comparison, TROPOMI retrievals were checked based on their quality assurance (QA) value. Only pixels with a QA value equal to or larger than 0.75 were selected, removing cloudy pixels (cloud radiance fraction > 0.5) and erroneous

retrievals (van Geffen et al., 2018). Note that all TROPOMI retrievals over the target scenes were compliant with the QA threshold.

In appendix A, tropospheric NO$_2$ VCD statistics for coincident TROPOMI and APEX pixels are provided for Flight #1 to #4 in Table 6 to 9, respectively. In total, 58 TROPOMI pixels were assessed. For each TROPOMI pixel acquired over the target area, the tropospheric NO$_2$ VCD is provided for both the TM5-MP-based (NO$_2$ VCD$_{TROPO}$) and CAMS-based (NO$_2$

VCD$_{TROPO-CRE}$) product. On average, NO$_2$ VCD$_{TROPO}$ is 6.8 x 10$^{15}$ molec cm$^{-2}$ and NO$_2$ VCD$_{TROPO-CRE}$ is 7.9 x 10$^{15}$ molec cm$^{-2}$. For the APEX NO$_2$ retrievals, the mean and median NO$_2$ VCD are provided for each TROPOMI pixel, as well as the standard deviation (SD), relative standard deviation or coefficient of variation (RSD) and minimum and maximum NO$_2$ VCD. On average over all flights, NO$_2$ VCD$_{APEX}$ is 8.0 x 10$^{15}$ molec cm$^{-2}$, which is in good agreement with the average CAMS-based TROPOMI NO$_2$ VCDs. The SD and RSD are on average 2.3 x 10$^{15}$ molec cm$^{-2}$ or 29%, respectively. They provide a measure

for the sub-pixel variability or spatial heterogeneity of the NO$_2$ field within a TROPOMI pixel, which is studied in more detail in Sect. 6. Highest concentrations are observed in the plume over the Antwerp harbour with maxima of up to 40 x 10$^{15}$ molec cm$^{-2}$.

Corresponding scatterplots and linear regression analyses of co-located TROPOMI and averaged APEX NO$_2$ VCD retrievals are provided in Fig. 8, for the ensemble of all four data sets. In Fig. 8a, TROPOMI pixels are only included in the

comparison when they are covered for more than 50% by APEX pixels, in order to reduce undersampling. However, for reference, linear regression analysis is also applied on all TROPOMI pixels having coincident APEX pixels and is provided in Fig. 8b. The data points are color-coded based on the number of APEX pixels averaged within a particular TROPOMI pixel. Vertical error bars indicate the overall error in NO$_2$ VCD$_{TROPO}$, while the horizontal whiskers represent the error in NO$_2$ VCD$_{APEX}$ retrievals, averaged over all APEX pixels coinciding with a particular TROPOMI pixel. Regression lines are color-

coded grey and black for the comparison of NO$_2$ VCD$_{APEX}$ with NO$_2$ VCD$_{TROPO}$, and NO$_2$ VCD$_{TROPO-CRE}$, respectively. Note that data points are shown for the comparison of NO$_2$ VCD$_{APEX}$ with VCD$_{TROPO-CRE}$ only. Corresponding correlation statistics are provided in Table 4 for each individual data set, as well as for the ensemble of the four data sets.



Overall for the ensemble of the four flights, a good agreement can be observed for both low and high retrievals. The standard TROPOMI $NO_2$ VCD product is well correlated (R= 0.92) and has a slope and intercept of 0.82 and 0.3 x $10^{15}$ molec $cm^{-2}$ with respect to APEX $NO_2$ reference observations. The observed negative bias is expected to be due to a combination of 1) the limited TROPOMI spatial resolution with respect to the occurring fine-scale gradients in polluted areas, and 2) the

limited spatial resolution of a priori input for the AMF computation, i.e. $NO_2$ profiles at 1° from the TM5-MP CTM and surface albedo at 0.5° from the OMI LER. When replacing the TM5-MP a priori $NO_2$ profiles by CAMS-based profiles, the correlation coefficient increases to 0.94 and the slope increases by 11% to 0.93. Correcting for the estimated systematic bias of 0.005 to 0.010 in the TROPOMI/OMI LER in case of clear-sky days, as discussed in Sect. 4.3.2, would scale up the TROPOMI VCD retrievals by 5 to 10%. In Fig. 8b, a less favorable slope (0.77) and a reduced correlation (R=0.86) can be observed due to the

effect of undersampling when considering all covered TROPOMI pixels. When considering only TROPOMI pixels which are fully covered by APEX observations (only 31 instead of 58 pixels), the statistics are of the same order as when applying the condition that TROPOMI pixels should be covered at least half by APEX observations. This assures us that the data set based on the latter condition is representative, while increasing the amount of TROPOMI pixels that can be assessed.

Note that Table 4 also shows correlation statistics for the comparison of the tropospheric $NO_2$ slant column product,

which has been compared in the same way as the VCDs. The slope is around 0.5 as the APEX airborne retrievals have a higher sensitivity to the $NO_2$ layer than the TROPOMI retrievals, resulting in larger SCDs. This is properly accounted for by the AMFs when converting to the vertical columns. When looking at the scatter, the SCDs exhibit a slightly larger correlation coefficient and lower root mean square error (RMSE), i.e. 7.8 and 8.1 x $10^{14}$ molec $cm^{-2}$, for the comparison of SCDs and VCDs, respectively. As discussed in Sect. 4.3.2, this could be related to noise introduced in the VCD retrieval by the difference

between the effective albedo variability and the albedo from the coarse OMI LER climatology, used in the computation of the AMFs.

The $NO_2$ VCD bias, defined by $VCD_{TROPO(-CRE)} - VCD_{APEX}$, and $NO_2$ VCD relative bias, defined by $(VCD_{TROPO(-CRE)} - VCD_{APEX}) / VCD_{APEX}$ x 100, has been calculated as well for the ensemble of the four data sets and is provided in Fig. 9 and Table 4. Data points and statistics are color-coded grey and black for the comparison of TM5-MP-based and CAMS-based

TROPOMI $NO_2$ VCDs with APEX $NO_2$ VCDs, respectively. On average, the bias is -1.2 ± 1.2 x $10^{15}$ molec $cm^{-2}$ or -14% ± 12% for the difference between the standard TROPOMI $NO_2$ VCD product and APEX. The bias is substantially reduced when replacing the coarse TM5-MP a priori $NO_2$ profiles by CAMS-based profiles, being -0.1 ± 1.0 x $10^{15}$ molec $cm^{-2}$ or -1% ± 12%. When the absolute value of the difference is taken, the bias is 1.3 x $10^{15}$ molec $cm^{-2}$ or 16%, and 0.7 x $10^{15}$ molec $cm^{-2}$ or 9% on average, when comparing $VCD_{APEX}$ with $VCD_{TROPO}$ and $VCD_{TROPO-CRE}$, respectively. In general a stronger bias can

be observed for high VCDs, related to the larger uncertainties on both the APEX and TROPOMI retrievals. Both sets of retrievals are well within the accuracy requirement of a maximum bias of 25-50% for the TROPOMI tropospheric $NO_2$ product for all individual compared pixels. These thresholds are indicated by the red dashed (25%) and full (50%) horizontal lines in Fig. 9b. Nevertheless, the standard tropospheric $NO_2$ product is clearly biased low over polluted areas when compared to





reference observations at higher resolution and this is consistent with the findings in other studies (Griffin et al., 2019; Ialongo et al., 2019; Zhao et al., 2019; Dimitropoulou et al., 2020; Verhoelst et al., 2020).

Remaining disagreements between the data sets can be potentially attributed to (1) different sensitivities to the $NO_2$ layer due to instrumental and algorithmic differences, and the a priori input in the radiative transfer simulations (studied in

Sect. 4.3), (2) differences in observation geometry and height, and (3) temporal differences in the observation of a dynamic $NO_2$ field. Concerning the latter point, APEX data was acquired over the target areas within one hour of the S-5P overpass time. Nevertheless, the potential impact of temporal $NO_2$ variability due to the time offset between the acquisition of the APEX and TROPOMI data sets has been investigated. In Fig. 10a, the same scatterplot and linear regression analysis is shown as in Fig. 8a, however, with the data points color-coded based on the absolute time offset between the TROPOMI overpass and the

mean acquisition time of APEX retrievals within the pixel. The data set does not exhibit a clear dependency on increasing time offset. In Fig. 10b, the observed $NO_2$ VCD bias, defined by $VCD_{TROPO-CRE}$ - $VCD_{APEX}$, has been plotted against the absolute time offset. The data set seems to be uncorrelated with a correlation coefficient of 0.02. Relatively low and high biases occur at both small and large time offsets, which is pointing at a low impact of the temporal $NO_2$ variability under the current conditions.

Both on 26 June and 29 June 2019, there were two early-afternoon S-5P overpasses over Belgium with a time difference between the two orbits of approximately 100 min. To assess the impact of the temporal $NO_2$ variability, the changes in the $NO_2$ field have been studied in the subsequent overpasses for the Belgian domain. On June 26, the absolute value of the differences observed is $3.8 \pm 5.3 \times 10^{14}$ molec $cm^{-2}$ or 12% $\pm$ 10%, on average. A maximum difference of $5.8 \times 10^{15}$ molec $cm^{-2}$ or 57% was observed for a pixel over the harbor of Antwerp, most likely due to a combination of moving air masses in the

key plumes and slight changes in the wind pattern. Additionally, the TROPOMI grids have different pixel sizes and orientations which also plays a role when acquiring and averaging the $NO_2$ patterns within the larger TROPOMI pixels. On June 29, the absolute value of the differences observed is $3.6 \pm 3.2 \times 10^{14}$ molec $cm^{-2}$ or 11% $\pm$ 8%, on average, with a maximum of $2.0 \times 10^{15}$ molec $cm^{-2}$, again seen over the harbour of Antwerp.

When analyzing the tropospheric $NO_2$ VCD diurnal variation, retrieved from the Uccle MAX-DOAS station (50.8° N,

4.4° E, 100 m a.s.l.) for the four campaign days, we see a low variability during the merged APEX flight time (11:00 - 13:44 UTC) for Flight #2 to Flight #4 (see Fig. 11). The relative standard deviation is lower than 10%. However, during Flight #1 we observe a strong increase of the VCD from 1.5 to $2.9 \times 10^{16}$ molec $cm^{-2}$ and a RSD of 32%. The instrument location is indicated by a white triangle in Fig. 7 and is pointed towards the Brussels city center (35° N).

## 6 Sub-pixel $NO_2$ variability and spatial smearing

Urbanised-industrialised areas are characterised by a strong spatial heterogeneity in the $NO_2$ field and steep spatial gradients. Current spaceborne observations typically have a resolution which is much coarser than the fine spatial structures in urban $NO_2$ plumes. The resulting smearing effect of the signal tends to bias the observed $NO_2$ field: urban cores are systematically



underestimated, while $NO_2$ is overestimated over urban background areas. Note that in this work urban background is defined as an area in a polluted environment, which is not directly affected by pollution plumes. The same can be observed over large industrial plumes that can extend over several tens of kilometers downwind of its source. When spaceborne observations are compared with ground-based station observations, such as PANDORA (Judd et al., 2019) and MAX-DOAS (Dimitropoulou

et al., 2020; Pinardi et al., 2020), the agreement is degraded with resolution as high concentrations in the pollution plumes are averaged out over a large area in the satellite data. Judd et al. (2019) downsample airborne GeoTASO VCDs (0.25 km x 0.25 km) to pseudo-TROPOMI (5 km x 5 km) and pseudo-OMI VCDs (18 km x 18 km). When compared to an ensemble of ten PANDORA stations, the initial $NO_2$ VCD correlation drops from 0.91 to 0.88 and 0.61, respectively, while the slope is reduced from 1.03 to 0.77 and 0.57, respectively.

The high-resolution APEX retrievals allow to monitor the effective variability in the $NO_2$ field at much finer scale than based on current and near-future spaceborne observations. One nadir TROPOMI pixel of 3.5 km x 7 km consists of approximately 2700 APEX observations. In case of fine-structured $NO_2$ plumes, the airborne data is expected to measure a larger variability, and stronger horizontal gradients, while we expect more blurring of the signal in the coarser TROPOMI data. This is illustrated in Fig. 12, based on a 15 km long southwest-northeast cross-section of the APEX and TROPOMI $NO_2$ VCD

grids, retrieved over Antwerp on 29 June 2019. APEX data shows considerably more spatial detail and observes higher columns over $NO_2$ hotspots when compared to TROPOMI. APEX measures peak $NO_2$ values which are 6 x $10^{15}$ molec cm$^{-2}$ or ~50% higher than seen by TROPOMI, while urban background pixels on their turn are overestimated up to 4 x $10^{15}$ molec cm$^{-2}$ or ~100% in the TROPOMI retrievals.

The SD and RSD, computed in Sect. 5.2.2 for coincident TROPOMI and APEX pixels, can be used as measures for the

sub-pixel variability or spatial heterogeneity of the $NO_2$ field within TROPOMI pixels, and are provided in appendix A. The RSD is obviously high for pixels that contain a steep gradient from urban background levels to $NO_2$ plume levels, e.g. pixel 7 in Flight #2, which has a $\mu \pm \sigma$ of $8.4 \pm 4.3$ x $10^{15}$ molec cm$^{-2}$ or a variability of ~51%. The variability is usually low when a pixel is entirely in the plume: e.g. for pixel 5 in Flight #2, $\mu \pm \sigma$ is $12.9 \pm 2.2$ x $10^{15}$ molec cm$^{-2}$ or a variability of ~17%. TROPOMI pixels classified as urban background, such as pixel 16 in Flight #2 can also exhibit considerable variability, with

a $\mu \pm \sigma$ of $5.0 \pm 2.0$ x $10^{15}$ molec cm$^{-2}$ or a variability of ~41%. This is due to high heterogeneity and the presence of small areas with moderate emissions, like a key road, industrial facility or small residential area. Note that in some conditions the (R)SD can be used as an indicator for the instrument precision. However, we assume that the data sets acquired over the cities do not contain areas where the $NO_2$ field is homogeneous enough to use it as a measure for the noise of the instrument.

## 6.1 Downsampling APEX to pseudo-TROPOMI $NO_2$ VCDs

In this section we investigate and quantify the impact of smearing of the effective signal due to the finite satellite pixel size. This is done based on the high-resolution APEX observations and under the assumption that the retrieved VCDs represent the true state of the $NO_2$ field. We have adopted a downsampling method described in Kim et al. (2016) and Judd et al. (2019):



First we construct a pseudo-TROPOMI VCD grid (VCD$_{pTROPO}$) by aggregating the APEX NO$_2$ VCDs (VCD$_{APEX}$) according to a weighted average technique within grid cells of 5 km x 5 km, 4.4 km x 4.4 km and 1 km x 1 km. The pixels are square in shape in order to avoid an orientation bias. Note that the original APEX VCD grid at 75 m x 120 m was regridded first to 100 m x 100 m for the same reason. The first two cases cover approximately the same area as a 7 km by 3.5 km (before 6 August

2019), and 5.5 km by 3.5 km (since 6 August 2019) TROPOMI nadir observation, respectively. The third case resembles a potential spatial resolution of future satellite or high-altitude pseudo-satellite (HAPS) missions, and is still a factor 10 larger than the APEX resolution. The respective NO$_2$ VCD grids are shown in Fig. 13 for the data set acquired over Antwerp on 29 June. At the resolution of 1 km x 1 km, different plumes can still be resolved and they can be largely linked to the key emission sources, such as the stacks in the harbour and the Antwerp ring road. However, at the resolution of 4.4 km x 4.4 km, and 5 km

x 5 km, only one merged plume can be distinguished downwind while it is not trivial to pinpoint its source(s). Note that the highest NO$_2$ levels are not observed for the pixels containing the sources, as NO$_2$ plumes are usually narrow and confined close to its source, resulting in a stronger smoothing effect for these pixels.

After regridding, the APEX NO$_2$ VCDs are subtracted from the pseudo-satellite VCDs. The resulting absolute and relative VCD differences allow to quantify the under- and overestimation bias in TROPOMI NO$_2$ retrievals over strongly

polluted regions, due to the smearing of the NO$_2$ signal. The approach allows to assess the impact solely related to the geometric effects resulting from the finite satellite pixel resolution. In the following, the approach is applied on two data sets acquired over Antwerp on 27 and 29 June 2019. The observed columns are larger for 29 June, while this day is also characterized by a larger variability.

In Fig. 14, the NO$_2$ VCD (relative) biases between pseudo-TROPOMI NO$_2$ VCDs at 4.4 km x 4.4 km and APEX NO$_2$

VCDs at 0.1 km x 0.1 km are illustrated for the data set acquired over Antwerp on 29 June 2019. Negative differences or red-coloured pixels point at an underestimation of NO$_2$ hotspots, while positive values or blue pixels point at overestimation of the urban background areas within the pseudo-TROPOMI VCDs. Whitish-coloured pixels represent no or very small bias. Obviously the strongest under- and overestimation appears over and in the vicinity of the main plumes, and more specifically over transition regions, and it is expected that the smoothing will be stronger when spatial gradients become stronger. Further

away from the patterns of enhanced NO$_2$, e.g. in the northeast and the south, the variability gets lower, resulting in a better agreement between the airborne high-resolution pixels and the relatively coarse pseudo-satellite pixels. However, the relative bias can still be substantial in the urban background areas due to the low NO$_2$ VCDs in the APEX data at the native resolution. The same behaviour was observed in Richter et al. (2005) when comparing NO$_2$ retrievals from GOME (40 km x 320 km) and SCIAMACHY (30 km x 60 km): while coincident observations agree very well over large areas of relatively homogeneous

NO$_2$ signals, they show considerable differences for areas with steep gradients in the NO$_2$ field.

Statistics, characterizing the NO$_2$ field, are provided for the two different APEX data sets acquired over Antwerp, in Table 5. The data set acquired on 27 June 2019 has a rather low mean VCD and variability of 7.6 ± 3.0 x 10$^{15}$ molec cm$^{-2}$, when compared to the data set acquired on 29 June 2019 (9.9 ± 5.4 x 10$^{15}$ molec cm$^{-2}$). Nevertheless, both areas represent an



urban $NO_2$ field characterized by relatively strong spatial gradients. Based on the study in Sect. 5.2.2, the sub-pixel variability can be up to 50% when covering a typical gradient in the urban $NO_2$ field.

For both data sets, statistics are provided as well for the computed pseudo-satellite VCDs at 5 km x 5 km, 4.4 km x 4.4 km, and 1 km x 1 km. When increasing the pixel size, the overall variability drops and the minima and maxima are less extreme due to the occurring smoothing.

In the last part of Table 5, statistics for the absolute value of the VCD differences are provided after subtracting the APEX $NO_2$ VCDs at 0.1 km x 0.1 km from the pseudo-satellite VCDs. The amount of under- or overestimation is around 1 to 2 x $10^{15}$ molec cm$^{-2}$ on average (5 km x 5 km grid), for the data set with relatively low (Flight #2) and high (Flight #4) urban $NO_2$ variability, respectively. The bias can be as high as 20 x $10^{15}$ molec cm$^{-2}$. The amount of under- or overestimation is still around 8%-10% on average for the pseudo-VCDs at 1 km x 1 km resolution. The difference seems, however, small between the two data sets acquired over Antwerp pointing out that 1) low or high variability is captured in more or less an equal way at the resolution of 1 km x 1 km and 2) this is a near-optimal resolution to capture strong urban emissions and associated gradients from space, at least under the current conditions. The bias increases with pixel resolution up to ~13% (Flight #2) and ~23% (Flight #4) for the gridsize at 5 km x 5 km. At this spatial resolution, the amount of variability in the data has clearly a stronger effect on the amount of smoothing of the effective signal. Maximum differences can be up to ~1900% and are due to the overestimation of retrievals with very low background values in the original APEX data (~0.3 x $10^{15}$ molec cm$^{-2}$). Based on a similar study applied on OMI data (13 km x 24 km) over the Contiguous United States (Kim et al., 2016), it was found that under- or overestimation biases are in the order of 5-10 x $10^{15}$ molec cm$^{-2}$ or 20-30% for major cities like Washington D.C. and New York. Biases are more than 100% for small-scale cities like Norfolk and Richmond. The stronger spatial smoothing observed in this study can be mainly explained by the coarser pixel resolution of OMI when compared to TROPOMI.

## 6.2 Simulations based on synthetic TROPOMI $NO_2$ VCDs

In Fig. 15, an approach is illustrated based on synthetic satellite $NO_2$ VCD data in order to study 1) the impact of spatial smoothing of the $NO_2$ signal, and 2) to which level spatial $NO_2$ features can still be resolved from space. Here satellite $NO_2$ VCDs are simulated assuming that they contain an isolated $NO_2$ hotspot of certain strength and size. The remaining part of the pixel is assigned a fixed value of 3 x $10^{15}$ molec cm$^{-2}$, representative for urban background. The $NO_2$ hotspots are defined by its relative size on the x-axis, expressed as the fraction of a 5.5 km by 3.5 km TROPOMI nadir pixel, and average hotspot $NO_2$ signal on the y-axis, ranging between 1 and 5 x $10^{16}$ molec cm$^{-2}$. In Fig. 15, the color-coded matrix values define the satellite $NO_2$ VCD based on a given $NO_2$ hotspot of certain size (x-axis) and strength (y-axis) within the satellite pixel. A threshold is used to identify whether or not a $NO_2$ signal within a TROPOMI pixel is still detectable, and is defined as the sum of the urban background VCD of 3 x $10^{15}$ molec cm$^{-2}$ and a $NO_2$ VCD detection limit of 2.1 x $10^{15}$ molec cm$^{-2}$, defined as three times the





TROPOMI theoretical precision requirement. The separation between the white area and synthetic $NO_2$ VCDs visualizes the threshold of 5.1 x $10^{15}$ molec $cm^{-2}$.

In case of a moderate source of 1 x $10^{16}$ molec $cm^{-2}$, the plume needs to fill 30% of the pixel, equivalent to 5.8 $km^2$, in order to be detectable, while in case of a strong source of 2.5 or 5 x $10^{16}$ molec $cm^{-2}$, the hotspot needs to fill only 10% (1.9 $km^2$) or 5% (1.0 $km^2$) of the pixel, respectively. Note that in case of a TROPOMI pixel size of 7 km by 3.5 km (product resolution at nadir until 6 August 2019), the size of the $NO_2$ hotspot needs to be 7.4 $km^2$, 2.3 $km^2$ and 1.2 $km^2$, respectively, in order to be detectable. In case a pixel is filled half by a $NO_2$ hotspot with a strength of 1 x $10^{16}$ molec $cm^{-2}$, its value will be 35% lower than the hotspot value due to smoothing, while this will be approximately 45% lower in case of a hotspot value of 5 x $10^{16}$ molec $cm^{-2}$.

## 7 Conclusions

Independent validation of the end-to-end mission performance is essential for the determination of S-5P data quality. It also provides critical information to identify and decide where and how to improve the overall data acquisition and processing chain. This is one of the first studies assessing TROPOMI tropospheric $NO_2$ retrievals over strongly polluted urban areas, based on the comparison with airborne high-resolution remote sensing observations. Satellite products can be optimally assessed with airborne data as a large amount of satellite pixels can be fully mapped at high resolution in a relatively short time interval, reducing the impact of mismatches in spatial and temporal representativeness. $NO_2$ VCDs retrieved from APEX, acquired on four consecutive clear-sky days (26-29 June 2019) over the cities of Brussels and Antwerp, have been compared with retrievals from coincident TROPOMI overpasses. On average a TROPOMI pixel has been fully covered by approximately 2700 APEX pixels and time differences between APEX and TROPOMI acquisitions were limited to less than one hour.

The comparison and assessment, discussed in Sect. 5, shows that the TROPOMI tropospheric $NO_2$ product meets the requirements in terms of precision and accuracy. Averaged over the four campaign days over Belgium, the precision of the TROPOMI $NO_2$ VCD product is estimated to be 5.6 ± 0.4 x $10^{14}$ molec $cm^{-2}$, thus within the targeted requirement of 7.0 x $10^{14}$ molec $cm^{-2}$. Overall for the ensemble of the four flights, the standard TROPOMI $NO_2$ VCD product is well correlated (R = 0.92) but biased negatively by -1.2 ± 1.2 x $10^{15}$ molec $cm^{-2}$ or -14% ± 12%, on average, with respect to coincident APEX $NO_2$ retrievals. When replacing the coarse 1° x 1° TM5-MP a priori $NO_2$ profiles by $NO_2$ profile shapes from the CAMS regional CTM ensemble at 0.1° x 0.1°, the slope increases by 11% to 0.93 and the bias is reduced to -0.1 ± 1.0 x $10^{15}$ molec $cm^{-2}$ or -1.0% ± 12%. When the absolute value of the difference is taken, the bias is 1.3 x $10^{15}$ molec $cm^{-2}$ or 16% and 0.7 x $10^{15}$ molec $cm^{-2}$ or 9% on average, when comparing APEX with TM5-MP-based and CAMS-based $NO_2$ VCDs, respectively. Both sets of retrievals are well within the TROPOMI accuracy requirement of a maximum bias of 25-50% for all individually compared pixels.

Nevertheless, TROPOMI is generally biased low over polluted areas when compared to ground-based or airborne observations and this is consistent with the findings in other studies, such as Griffin et al. (2019), Ialongo et al. (2019), Zhao





et al. (2019), Dimitropoulou et al. (2020), and Verhoelst et al. (2020). This is largely due to a combination of 1) the limited spatial resolution of TROPOMI with respect to the strong $NO_2$ horizontal and vertical gradients, 2) the limited spatial resolution of a priori input for the AMF computation, i.e. $NO_2$ profiles at 1° from the TM5-MP CTM and surface albedo at 0.5° from the OMI LER, and 3) the estimated bias of 0.005-0.010 in the TROPOMI/OMI LER. Since 6 August 2019, the spatial resolution

is upgraded from 3.5 km x 7 km at nadir observations to 3.5 km x 5.5 km. The $NO_2$ product could be further improved for retrievals over polluted regions by making use of 1) a priori $NO_2$ profiles from a high-resolution CTM, if available, such as the CAMS-regional ensemble at 0.1° and 2) an improved albedo product. A G3_LER daily map product at 0.1°, directly retrieved from the TROPOMI L1B radiances, is currently under development. Furthermore, a surface albedo adjustment scheme will become operational after reprocessing the L1B product to v2, planned for the second half of 2020.

The TROPOMI spatial resolution is limited to resolve fine-scale urban $NO_2$ plumes and can cause a considerable smoothing effect in case of the observation of strongly polluted scenes with steep gradients. The high-resolution APEX retrievals allow to monitor the effective horizontal variability in the $NO_2$ field at much finer scale. In Sect. 6 the impact of smearing of the effective signal due to the finite satellite pixel size was studied based on a downsampling approach of the APEX retrievals. Assuming a pixel size of 25 to 20 $km^2$, equivalent to the initial 3.5 km x 7 km and new TROPOMI 3.5 km x

5.5 km spatial resolution (at nadir), the amount of underestimation of peak plume values and overestimation of urban background in the TROPOMI data is expected to be in the order of 1-2 x $10^{15}$ molec $cm^{-2}$, on average, or 10% - 20%, depending on the amount of heterogeneity in the $NO_2$ field. The average under- and overestimation is further reduced to 0.6-0.9 x $10^{15}$ molec $cm^{-2}$, or smaller than 10%, when increasing the pixel size to 1 $km^2$. Therefore, detailed air quality studies at the city scale still require observations at higher spatial resolution, in the order of 1 $km^2$ or better, in order to resolve all fine-scale

structures within the typical heterogeneous $NO_2$ field.

A validation strategy for TROPOMI tropospheric $NO_2$ retrievals has been presented based on airborne mapping data, which can be valuable for future assessments of S-5P and upcoming satellite missions, such as S-5, S-4, GEMS and TEMPO. The main focus was to quantify the retrieval uncertainties in polluted regions and results from the comparison with independent airborne APEX retrievals have shown that the TROPOMI tropospheric $NO_2$ product meet the requirements in terms of accuracy

and precision. However, further validation studies are required, focusing on more sites with different geophysical properties and varying pollution levels, including background areas, extreme albedo sites, other seasons, and cloudy scenes, among others, in order to assess as well the performance in suchlike conditions.

*Data availability.* The data are available upon request to the corresponding author.

Author contributions. FT undertook the validation study and writing of the manuscript under supervision of MVR. FT, AM, BB, and MVR planned and organised the measurement campaign. MDI, GP, and ED contributed to the campaign. MDI and BB preprocessed the APEX spectra. HE processed the customized TROPOMI tropospheric $NO_2$ product based on CAMS a



priori profiles. FT performed the APEX VCD retrievals and comparison study. AM, MDI, GP, ED, HE, and MVR contributed in scientific discussions. All co-authors reviewed, discussed the results and commented on the final manuscript.

*Competing interests.* The authors declare that they have no conflict of interest.

*Acknowledgements.* The Belgian Federal Science Policy Office is gratefully appreciated for funding the APEX aircraft activities in the framework of the STEREO programme. The European Space Agency is gratefully acknowledged for funding the S-5P Campaign Implementation Plan (Tack et al., 2018) and other TROPOMI retrieval and validation activities at BIRA. The authors wish to thank Frans Fierens and Charlotte Vanpoucke from the Belgian Interregional Environment Agency for

providing emission inventory data. We also wish to thank Ben Somers from the KU Leuven University for providing the Microtops handheld sun photometer and our colleagues François Hendrick and Martina Friedrich from BIRA for helping to conduct DOAS measurements during the S5PVAL-BE campaign.

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

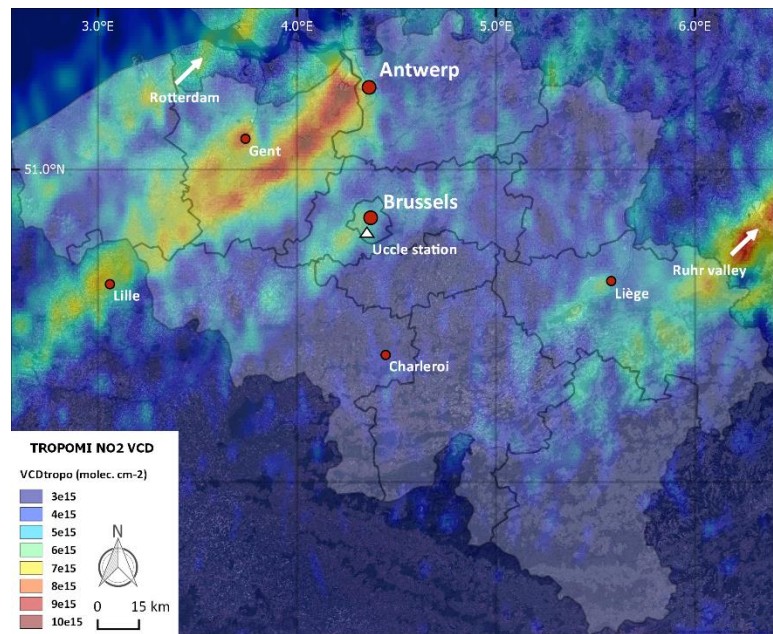

**Figure 1.** Tropospheric NO$_2$ hotspots observed over Belgium by TROPOMI, based on an early afternoon S-5P orbit (8826) on 27 June 2019 (OFFL v1.03.01 – thin plate spline interpolation at 0.01°) (© Google Maps). Red markers indicate the five largest Belgian cities. The white triangle indicates the location of the Uccle station (50.8° N, 4.4° E, 100 m a.s.l.). White arrows indicate the source locations for long-range transport plumes over Belgium. On 27 June 2019, there was a northeasterly wind (36°).

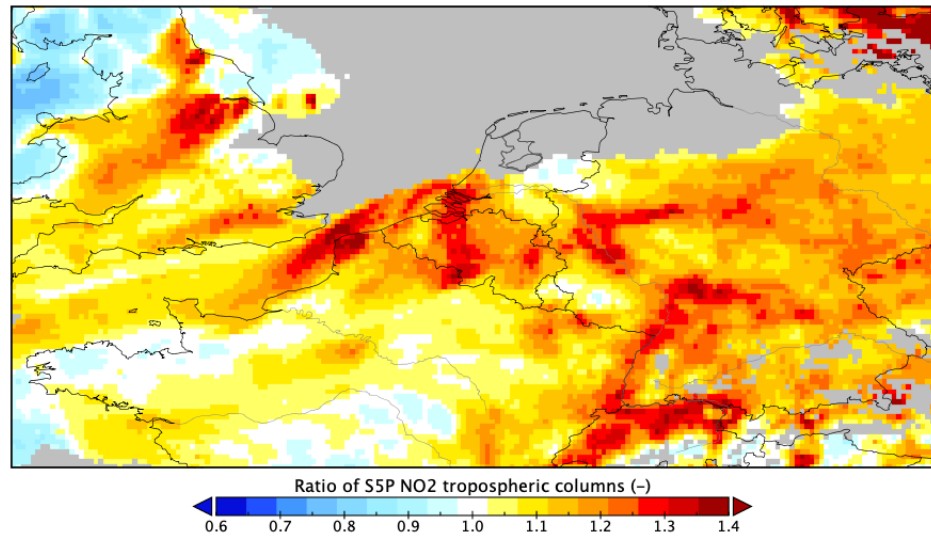

**Figure 2.** Ratio of TROPOMI tropospheric $NO_2$ columns when using the CAMS-regional a priori $NO_2$ profiles with respect to TM5-MP a priori profiles over Belgium and neighboring countries on 27 June 2019 (orbit 8826).



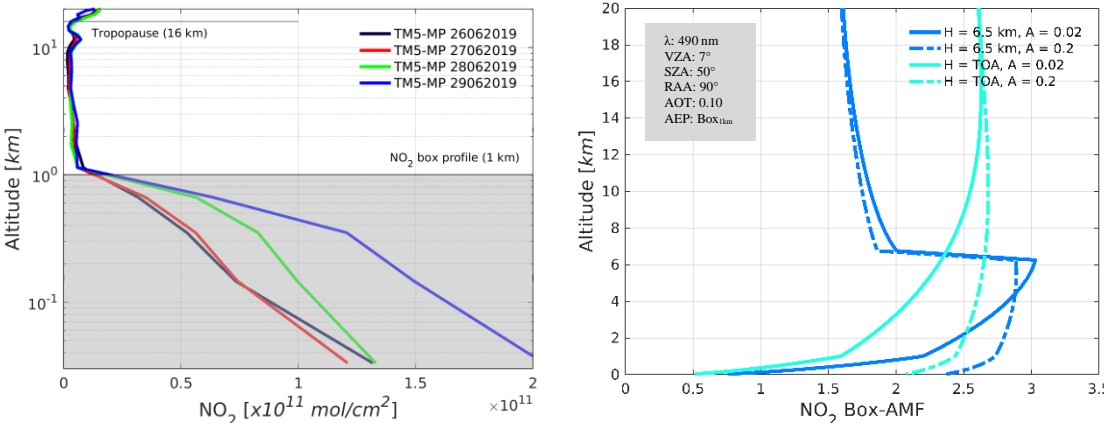

**Figure 3. (a)** Representation of a well-mixed $NO_2$ box profile of 1 km thickness and TM5-MP $NO_2$ profiles interpolated over the campaign sites for Flight #1 to #4, and **(b)** height-dependent box AMFs representing the vertical sensitivity to $NO_2$, illustrated for APEX, operating at 6.5 km a.g.l., and TROPOMI, for both a low and high surface reflectance scenario.





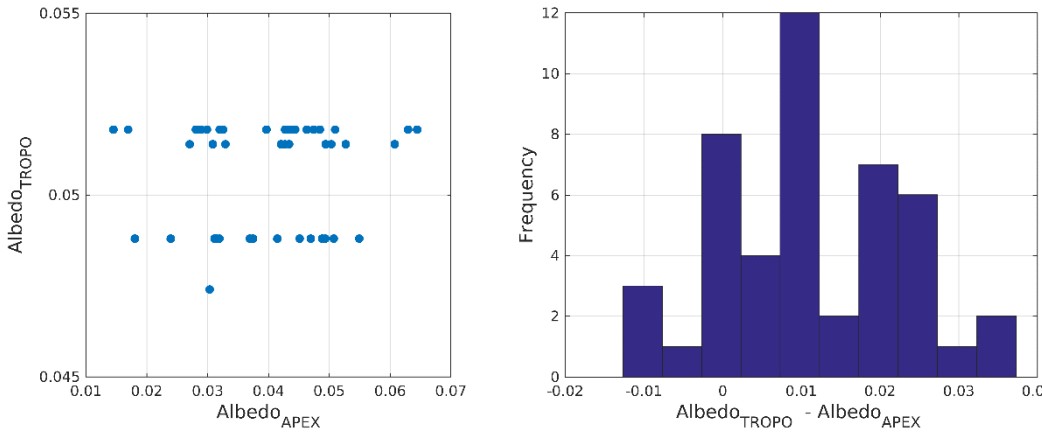

**Figure 4. (a)** Scatterplot and **(b)** histogram for the comparison between TROPOMI and APEX albedo for the ensemble of the four APEX data sets.





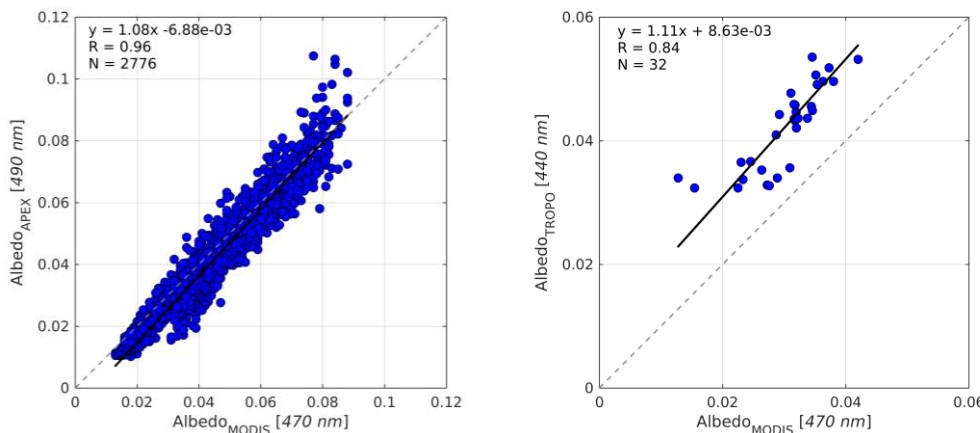

**Figure 5.** Scatterplots and linear regression analyses of co-located **(a)** APEX and MODIS, and **(b)** TROPOMI and MODIS albedo pixels for 27 June 2019.



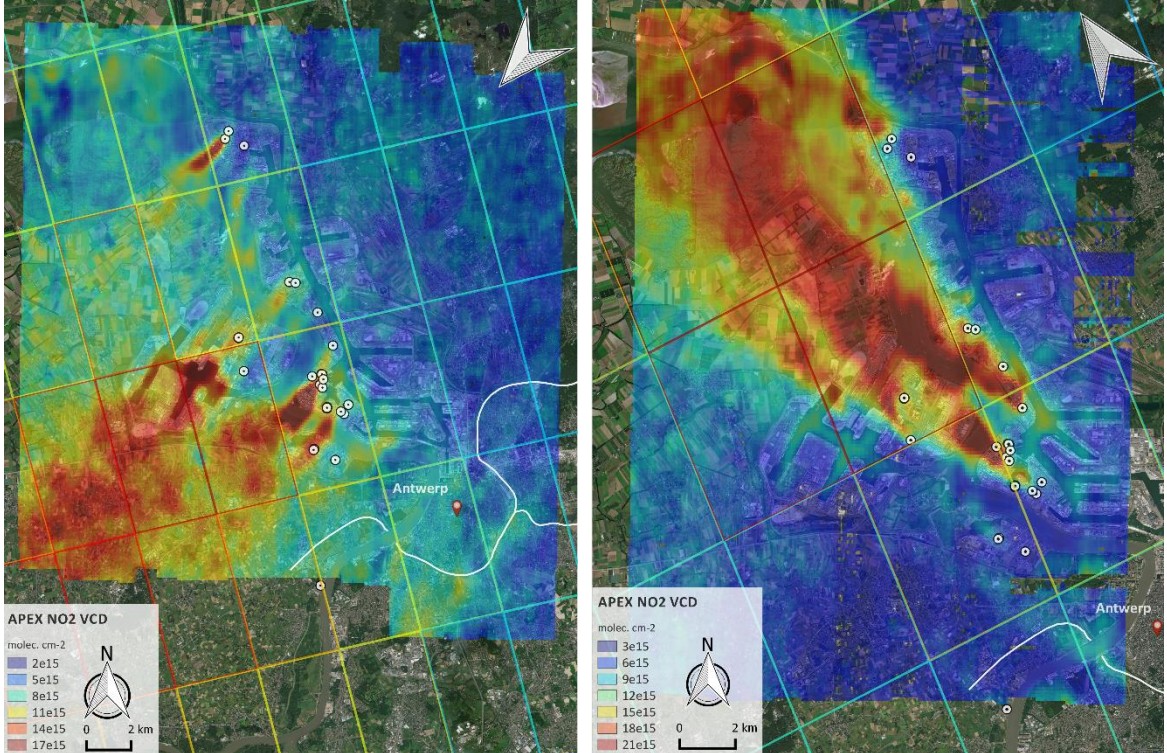

**Figure 6.** Tropospheric NO₂ VCD grids retrieved over Antwerp on **(a)** 27 June (Flight #2) and **(b)** 29 June (Flight #4), 2019. Note that different color scales were applied in order to optimize the dynamic range of each data set. White dots indicate the point sources, emitting more than 10 kg of NOx per hour, according to the emission inventory (2017) of the Belgian Interregional Environment Agency. Line sources such as the key highways and city ring road are indicated by white lines. Coinciding TROPOMI tropospheric NO₂ VCD retrievals are overlayed as color-coded polygons. White wind vectors indicate the surface wind, averaged over the APEX acquisition time, as provided in Table 1 (© Google Maps).

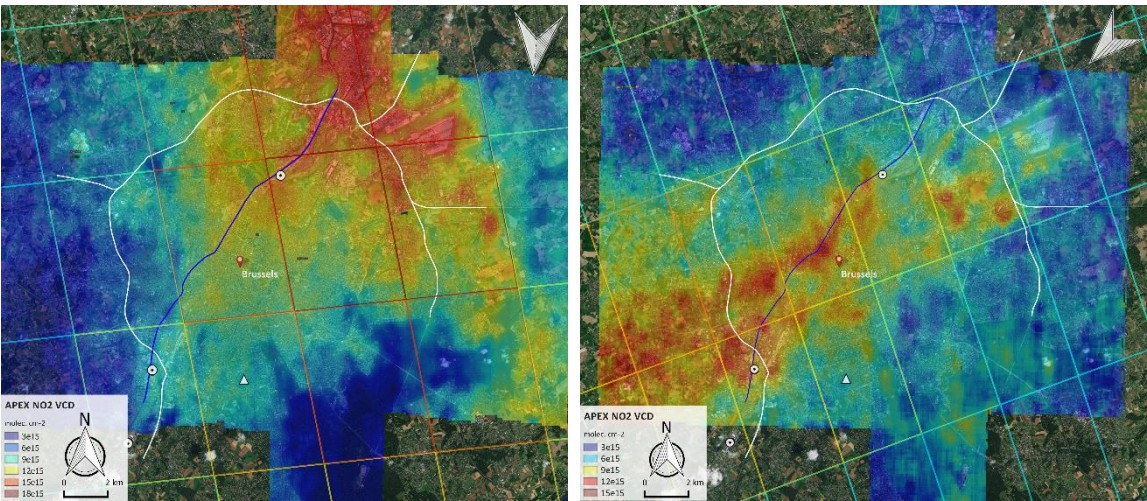

**Figure 7.** Tropospheric NO₂ VCD grids retrieved over Brussels on **(a)** 26 June (Flight #1) and **(b)** 28 June (Flight #3), 2019. Note that different color scales were applied in order to optimize the dynamic range of each data set. White dots indicate the point sources, emitting more than 10 kg of NOx per hour, according to the emission inventory (2017) of the Belgian Interregional Environment Agency. The white triangle indicates the location of the Uccle MAX-DOAS station. Line sources such as the key highways and city ring road are indicated by white lines. Coinciding TROPOMI tropospheric NO₂ VCD retrievals are overlayed as color-coded polygons. White wind vectors indicate the surface wind, averaged over the APEX acquisition time, as provided in Table 1 (© Google Maps).



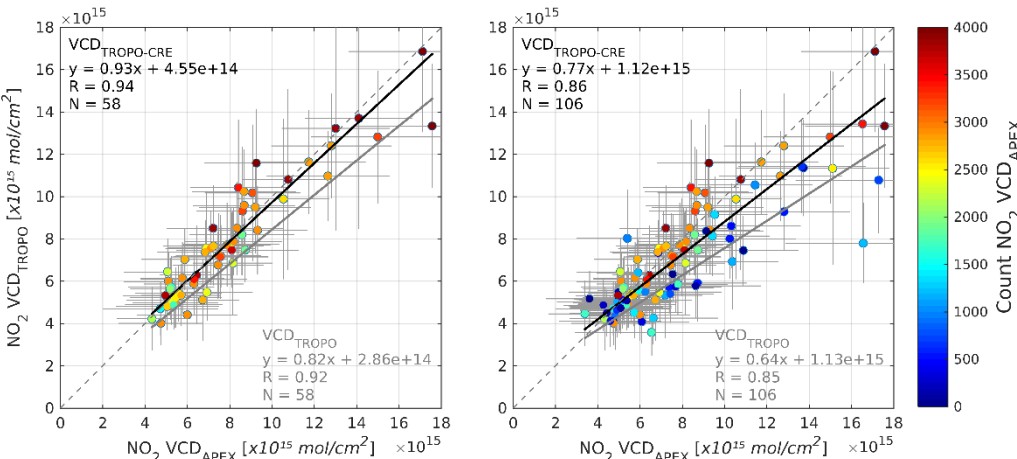

**Figure 8.** Scatterplots and linear regression analyses of co-located TROPOMI and averaged APEX $NO_2$ VCD retrievals for the data sets acquired on 26-29 June 2019. Regression lines and statistics are color-coded grey and black for the comparison of $NO_2$ $VCD_{APEX}$ with $NO_2$ $VCD_{TROPO}$ and $VCD_{TROPO-CRE}$, respectively. Note that data points are shown for the comparison of $NO_2$ $VCD_{APEX}$ with $VCD_{TROPO-CRE}$ only. Vertical error bars indicate the overall errors in $NO_2$ $VCD_{TROPO}$, while the horizontal whiskers represent the errors in $NO_2$ $VCD_{APEX}$ retrievals, averaged over all APEX pixels within the footprint of a co-located TROPOMI pixel. Data points are color-coded based on the number of APEX pixels averaged within a TROPOMI pixel. In **(a)**, TROPOMI pixels are only included in the comparison when they are covered for more than 50% by APEX pixels in order to avoid undersampling, while in **(b)**, as a reference, all TROPOMI pixels having coincident APEX pixels are analysed.



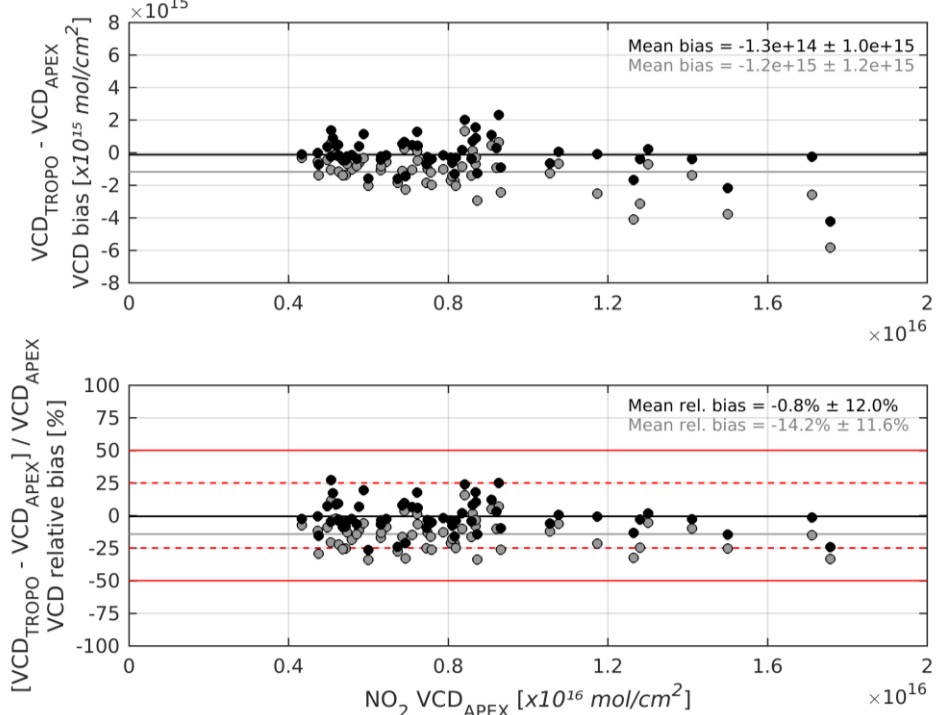

**Figure 9. (a)** NO₂ VCD bias (VCD$_{TROPO(-CRE)}$ – VCD$_{APEX}$) and **(b)** NO₂ VCD relative bias ((VCD$_{TROPO(-CRE)}$ – VCD$_{APEX}$)/ VCD$_{APEX}$ x 100) for the ensemble of the four data sets, acquired during the S5PVAL-BE campaign. Data points and statistics are color-coded grey and black for the comparison of TM5-MP-based, and CAMS-based TROPOMI VCD retrievals with APEX, respectively, in analogy to Fig. 8. The grey and black horizontal lines represent the average (relative) bias. The red dashed and full horizontal lines represent the 25% and 50% bias between coincident TROPOMI and APEX NO₂ VCDs, respectively.





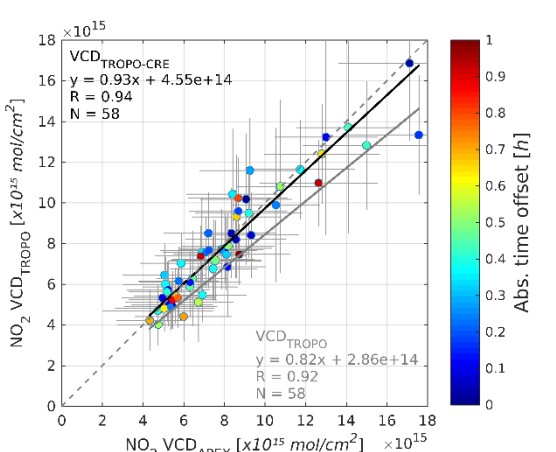

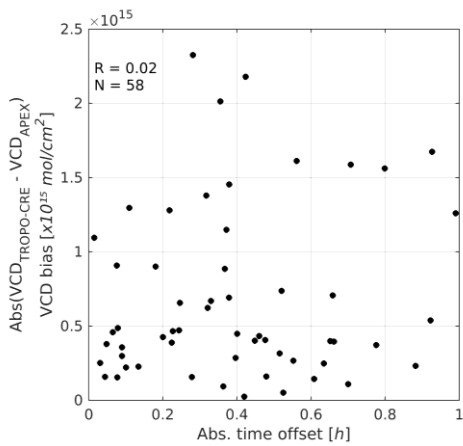

**Figure 10. (a)** Same as Fig. 8.a, but data points are color-coded based on the absolute time offset between TROPOMI overpass and mean acquisition time of APEX retrievals within the TROPOMI pixel, and in **(b)** the observed $NO_2$ VCD bias, defined by $VCD_{TROPO-CRE} - VCD_{APEX}$, has been plotted against the absolute time offset.



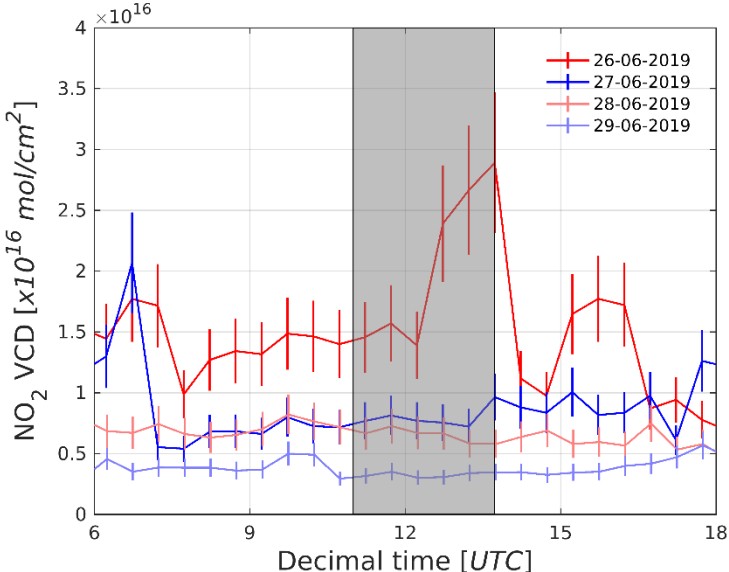

**Figure 11.** Tropospheric NO$_2$ VCD diurnal variation between 80° sunrise and sunset, retrieved from the Uccle MAX-DOAS station on 26-29 June 2019. The instrument is pointed towards the Brussels city center (35° N). Vertical error bars indicate the NO$_2$ VCD error for each retrieval. The grey zone indicates the merged APEX flight time (11:00 - 13:44 UTC) for 26 to 29 June.





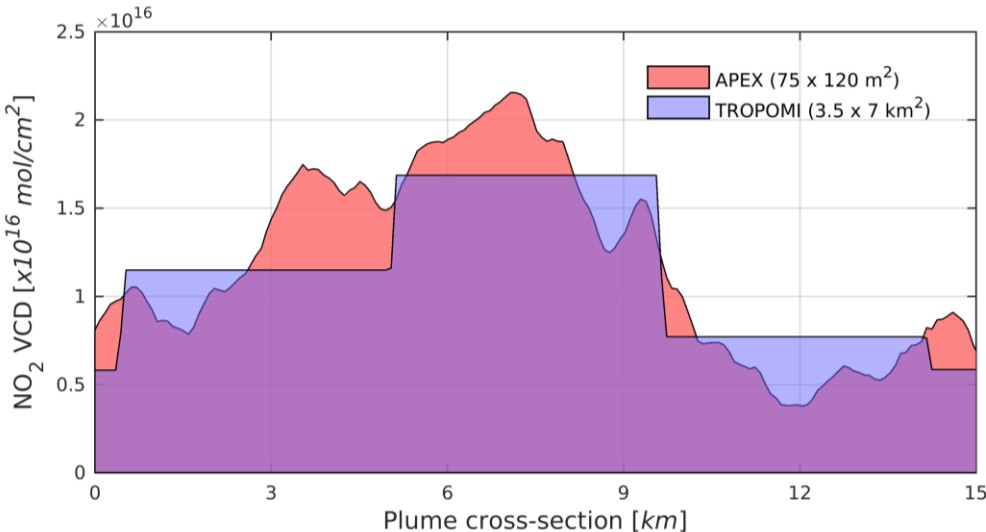

**Figure 12.** APEX and TROPOMI NO₂ VCDs along a southwest-northeast 15 km long cross-section taken perpendicular to the major NO₂ plume retrieved over Antwerp on 29 June 2019. Approximately five TROPOMI pixels and 150 APEX pixels are sampled.

(a) NO$_2$ VCD$_{APEX}$ 0.1 km x 0.1 km    (b) NO$_2$ VCD$_{pTROPO}$ 1 km x 1 km

(c) NO$_2$ VCD$_{pTROPO}$ 4.4 km x 4.4 km    (d) NO$_2$ VCD$_{pTROPO}$ 5 km x 5 km

**Figure 13**. **(a)** APEX NO$_2$ VCD grid retrieved over Antwerp on 29 June, at 0.1 km x 0.1 km resolution, and the computed pseudo-satellite NO$_2$ VCDs grids at **(b)** 1 km x 1 km, **(c)** 4.4 km x 4.4 km, and **(d)** 5 km x 5 km, respectively. In the 4 plots, the same color-code is applied with the legend provided in (a) (© Google Maps).

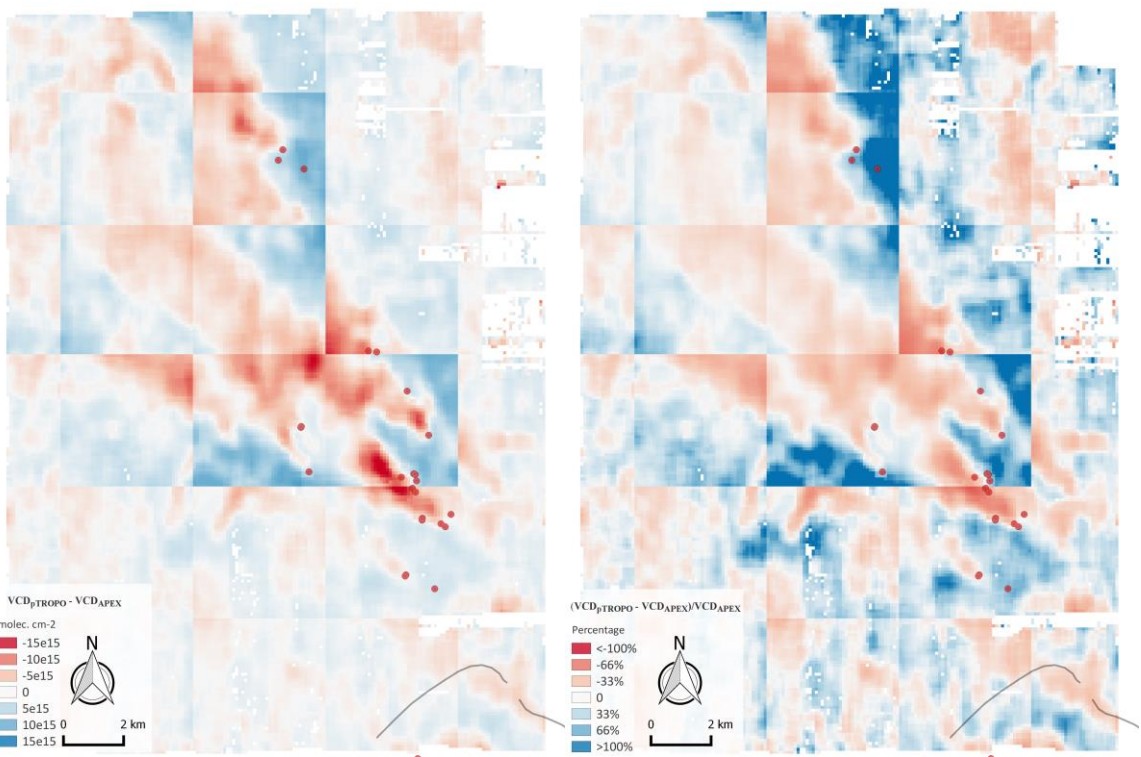

**Figure 14. (a)** The NO$_2$ VCD bias (VCD$_{pTROPO}$ – VCD$_{APEX}$) and **(b)** relative bias ((VCD$_{pTROPO}$ – VCD$_{APEX}$)/ VCD$_{APEX}$ x 100) for the APEX data set acquired over Antwerp on 29 June 2019. VCD$_{pTROPO}$ are pseudo-TROPOMI NO$_2$ VCDs, constructed by averaging the APEX NO$_2$ VCDs within grid cells of 4.4 km x 4.4 km.



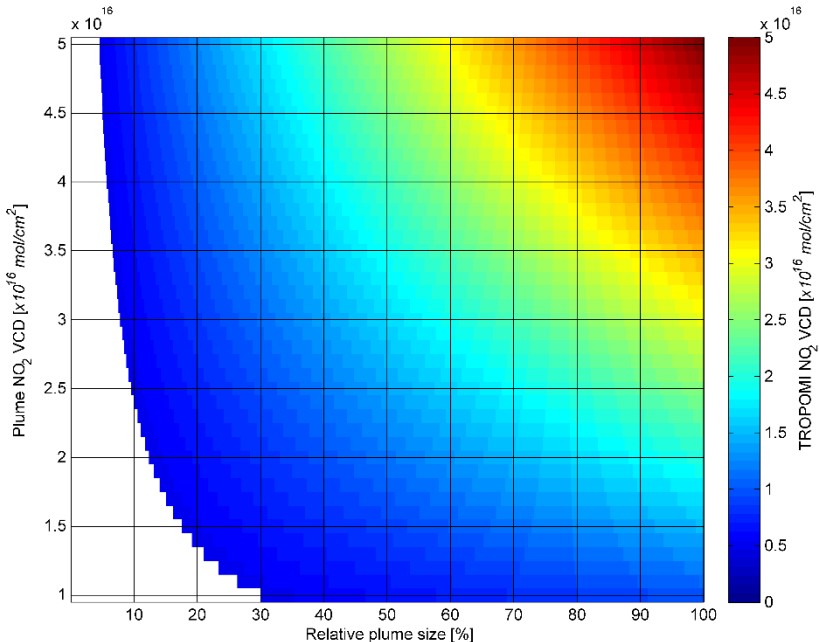

**Figure 15.** Simulations of NO₂ VCDs based on an isolated NO₂ hotspot surrounded by urban background pixels ($3 \times 10^{15}$ molec cm⁻²). The NO₂ hotspot is defined by its relative size on the x-axis, expressed as the fraction of a 3.5 km by 5.5 km TROPOMI nadir pixel, and average NO₂ signal strength on the y-axis. The separation between the white area and synthetic NO₂ VCDs corresponds to the hotspot detection threshold of $5.1 \times 10^{15}$ molec cm⁻².



**Table 1.** Mapping flights characteristics, and meteorological and environmental conditions for the four APEX flights, acquired over the cities of Antwerp and Brussels, in the framework of the S5PVAL-BE campaign.

| | Flight #1 | Flight #2 | Flight #3 | Flight #4 |
|---|---|---|---|---|
| Site[a] | Brussels | Antwerp | Brussels | Antwerp |
| Date | 26-06-2019 | 27-06-2019 | 28-06-2019 | 29-06-2019 |
| Day of year / week | 177 / Wednesday | 178 / Thursday | 179 / Friday | 180 / Saturday |
| Flight time LT (UTC+2) | 14:07–15:44 | 13:37–15:23 | 13:52–15:26 | 13:00–14:34 |
| TROPOMI overpass LT (UTC+2) | 13:16 (orbit 08811) 14:56 (orbit 08812) | 14:37 (orbit 08826) | 14:19 (orbit 08840) | 14:00 (orbit 08854) 15:41 (orbit 08855) |
| # flight lines | 12 | 11 | 12 | 11 |
| Flight pattern (Heading ) | 0°, 180° | 0°, 180° | 0°, 180° | 0°, 180° |
| SZA | 28°–36° | 28°–34° | 28°–34° | 29°–30° |
| Average wind direction | 4° | 36° | 49° | 143° |
| Average wind speed | 3.7 m s$^{-1}$ | 3.7 m s$^{-1}$ | 2.6 m s$^{-1}$ | 2.6 m s$^{-1}$ |
| Average temperature | 26° C | 23° C | 24° C | 30° C |
| Average PBL height | 684 m | 888 m | 798 m | No Data |
| Average AOT (440 nm) | 0.57 | 0.16 | 0.15 | 0.09 |
| Average AOT (500 nm) | 0.51 | 0.15 | 0.15 | 0.10 |
| Lat / Long | 50.8° N / 4.4° E | 51.2° N / 4.4° E | 50.8° N / 4.4° E | 51.2° N / 4.4° E |
| Average terrain altitude (a.s.l.) | 76 m | 10 m | 76 m | 10 m |

[a] Wind and temperature data are collected from weather stations of the Royal Meteorological Institute of Belgium (RMI), i.e. Uccle station (50.8° N, 4.4° E, 100 m a.s.l.) for Brussels, and Stabroek station (51.3° N, 4.4° E, 4 m a.s.l.) for Antwerp and measurements are averaged over the time of flight. PBL height was obtained from the backscatter profiles of a Vaisala CL51 ALC ceilometer operated by RMI in Uccle. The aerosol optical thickness (AOT level 1.5) was measured by the CIMEL AERONET station (Holben et al., 1998) in Uccle.



**Table 2.** TROPOMI and APEX specifications for the S5PVAL-BE campaign, defined for APEX for a nominal altitude of 6.5 km a.g.l. Spectrometer characteristics are provided for the APEX VNIR detector and the TROPOMI UV-VIS channel only. The effective APEX spatial resolution is provided after applying spatial aggregation of the spectra for signal-to-noise enhancement.

| | TROPOMI (UV-VIS) | APEX (VNIR) |
|---|---|---|
| Orbit | Polar, sun-synchronous | - |
| Temporal resolution | Daily global coverage (13:30 local solar time) | - |
| Wavelength range | 305–499 nm | 370–970 nm |
| Spectral resolution (FWHM) | 0.45–0.65 nm | 0.9–3.2 nm |
| FOV across-track | 108° | 28° |
| IFOV across-track | 0.24° | 0.028° |
| Flight altitude | 824 km | 6.5 km |
| Swath width | 2600 km | 3.2 km |
| Ground speed | 7800 m s$^{-1}$ | 72 m s$^{-1}$ |
| Across-track spatial resolution (nadir) | 3500 m | 75 m |
| Along-track spatial resolution (nadir) | 7000 m[a] | 120 m |
| Signal-to-noise ratio | 800-1000 | 2500 |
| NO$_2$ SCD detection limit (molec cm$^{-2}$) | ~5.6 x 10$^{14}$ | ~2.6 x 10$^{15}$ |
| Temperature stabilisation | Yes | Yes |
| Radiometric calibration | Yes | Yes |
| Weight | 220 kg | 354 kg |
| Size (LxWxH) | 0.75x0.56x1.4 m$^3$ | 0.83x0.64x0.56 m$^3$ |
| Power consumption | 170 W | 2100 W |
| Scanning | Pushbroom | Pushbroom |

[a] 5500 m since 6 August 2019. This scenario has been successfully tested during the S-5P Commissioning Phase and it was recommended during the In-Orbit Commissioning Review (IOCR) to be implemented during the operational phase.





**Table 3.** Overview of the key parameters for the DOAS spectral fitting and $NO_2$ slant column retrieval.

| | $NO_2$ $VCD_{TROPO}$ | $NO_2$ $VCD_{APEX}$ |
|---|---|---|
| $\lambda$ calibration | Solar irradiance and earthshine radiance | Solar spectrum (Chance and Kurucz, 2010) |
| Spectral fitting code | TROPOMI DOAS software based on optimal estimation solver (van Geffen et al., 2018) | QDOAS (Fayt et al., 2016) |
| Fitting interval | 405–465 nm | 470–510 nm |
| Cross-sections | | |
| $NO_2$ | Vandaele et al. (1998), at 220K | Vandaele et al. (1998), at 294K |
| $O_3$ | Gorshelev et al. (2014) and Serdyuchenko et al. (2014), at 243 K | n/a |
| $O_4$ | Thalman and Volkamer (2013), at 293 K | Thalman and Volkamer (2013), at 293 K |
| $H_2O_{vap}$ | HITRAN 2012 (van Geffen et al., 2015) | n/a |
| $H_2O_{liq}$ | Pope and Fry (1997) | n/a |
| Ring effect | Chance and Spurr (1997) | Chance and Spurr (1997) |
| Polynomial term | Order 5 | Order 5 |





**Table 4.** Correlation statistics between coincident APEX and TROPOMI $NO_2$ SCD and VCD products (OFFL v1.03.01) for the different flights. The last row "All data" considers all four data sets together. TROPOMI pixels are only compared with the average of all APEX pixels within the footprint, when they are covered for more than 50% by APEX pixels. The $NO_2$ VCD bias is defined by $VCD_{TROPO(-CRE)} - VCD_{APEX}$ and $NO_2$ VCD relative bias is defined by $(VCD_{TROPO(-CRE)} - VCD_{APEX}) / VCD_{APEX} \times 100$. Alpha (α) and beta (β) are the intercept and slope of the linear regression fit.

| | | $NO_2$ $SCD_{TROPO}$ vs $SCD_{APEX}$ | | | $NO_2$ $VCD_{TROPO}$ vs $VCD_{APEX}$ | | | | | $NO_2$ $VCD_{TROPO-CRE}$ vs $VCD_{APEX}$ | | | | |
| --- | --- | --- | --- | --- | --- | --- | --- | --- | --- | --- | --- | --- | --- |
| | N | R | β | α x10^15 | R | β | α x10^15 | Bias x10^15 | Bias % | R | β | α x10^15 | Bias x10^15 | Bias % |
| **Flight #1** (orbit 08812) | 12 | 0.96 | 0.66 | -0.67 | 0.94 | 0.98 | -0.40 | -0.54 | -6.1 | 0.94 | 1.08 | -0.64 | 0.04 | 0.2 |
| **Flight #2** (orbit 08826) | 21 | 0.95 | 0.43 | 0.68 | 0.95 | 0.70 | 0.64 | -1.63 | -20.8 | 0.95 | 0.94 | 0.30 | -0.15 | -1.5 |
| **Flight #3** (orbit 08840) | 15 | 0.93 | 0.52 | -0.15 | 0.92 | 0.93 | -0.26 | -0.73 | -10.5 | 0.91 | 1.11 | -0.70 | 0.04 | 0.8 |
| **Flight #4** (orbit 08854) | 10 | 0.94 | 0.45 | 1.03 | 0.93 | 0.71 | 1.13 | -1.77 | -15.4 | 0.93 | 0.83 | 1.18 | -0.54 | -2.8 |
| **All data** | 58 | 0.94 | 0.46 | 0.46 | 0.92 | 0.82 | 0.29 | -1.20 | -14.2 | 0.94 | 0.93 | 0.46 | -0.13 | -0.8 |





**Table 5.** NO$_2$ VCD statistics for (1) two different APEX data sets acquired over Antwerp on 27 and 29 June 2019, (2) pseudo-TROPOMI grids (5 km x 5 km, 4.4 km x 4.4 km, and 1 km x 1 km) constructed by aggregating the native APEX NO$_2$ VCDs from both former data sets, and (3) absolute and relative differences between the constructed pseudo-TROPOMI NO$_2$ VCDs and original APEX VCDs.

| | Antwerp Flight #2 (27-06-2019) | | | Antwerp Flight #4 (29-06-2019) | | |
|---|---|---|---|---|---|---|
| **NO$_2$ VCD$_{APEX}$** | **0.1 x 0.1 km$^2$** | | | **0.1 x 0.1 km$^2$** | | |
| Mean (x 10$^{15}$ molec cm$^{-2}$) | 7.6 | | | 9.9 | | |
| SD (x 10$^{15}$ molec cm$^{-2}$) | 3.0 | | | 5.4 | | |
| Min (x 10$^{15}$ molec cm$^{-2}$) | 0.3 | | | 1.5 | | |
| Max (x 10$^{15}$ molec cm$^{-2}$) | 27.4 | | | 32.7 | | |
| **NO$_2$ VCD$_{pTROPO}$** | **5 x 5 km$^2$** | **4.4 x 4.4 km$^2$** | **1 x 1 km$^2$** | **5 x 5 km$^2$** | **4.4 x 4.4 km$^2$** | **1 x 1 km$^2$** |
| Mean (x 10$^{15}$ molec cm$^{-2}$) | 7.6 | 7.6 | 7.6 | 9.9 | 9.9 | 9.9 |
| SD (x 10$^{15}$ molec cm$^{-2}$) | 2.6 | 2.7 | 2.8 | 4.4 | 4.6 | 5.2 |
| Min (x 10$^{15}$ molec cm$^{-2}$) | 4.6 | 4.5 | 3.4 | 4.4 | 3.8 | 3.6 |
| Max (x 10$^{15}$ molec cm$^{-2}$) | 13.9 | 14.0 | 20.1 | 18.2 | 18.6 | 24.2 |
| **Abs(VCD$_{pTROPO}$ − VCD$_{APEX}$)** | **5 x 5 km$^2$** | **4.4 x 4.4 km$^2$** | **1 x 1 km$^2$** | **5 x 5 km$^2$** | **4.4 x 4.4 km$^2$** | **1 x 1 km$^2$** |
| Mean (x 10$^{15}$ molec cm$^{-2}$) | 1.0 | 0.9 | 0.6 | 2.0 | 1.8 | 0.9 |
| SD (x 10$^{15}$ molec cm$^{-2}$) | 1.1 | 1.0 | 0.6 | 2.2 | 2.0 | 1.0 |
| Max (x 10$^{15}$ molec cm$^{-2}$) | 16.9 | 16.0 | 14.6 | 19.7 | 19.2 | 17.5 |
| Mean (%) | 13 | 13 | 8 | 23 | 21 | 10 |
| SD (%) | 15 | 14 | 9 | 29 | 25 | 11 |
| Max (%) | 1887 | 1759 | 1104 | 352 | 342 | 235 |





## Appendix A: Tropospheric NO₂ VCD statistics for coincident TROPOMI and APEX pixels

**Table 6.** Tropospheric NO₂ VCD statistics for coincident TROPOMI and APEX pixels for Flight #1, orbit 08812. APEX statistics are computed for all TROPOMI pixels covered by more than 50% by APEX pixels.

| NO₂ VCD$_{TROPO}$ (x 10$^{15}$ molec cm$^{-2}$) | | | NO₂ VCD$_{APEX}$ (x 10$^{15}$ molec cm$^{-2}$) | | | | | | |
|---|---|---|---|---|---|---|---|---|---|
| Pixel ID | VCD$_{TROPO}$ | VCD$_{TROPO-CRE}$ | Count | Mean | Median | SD | RSD(%)[a] | Min[b] | Max[b] |
| 1 | 7.1 | 7.6 | 2477 | 6.9 | 6.9 | 1.9 | 26.9 | 3.2 | 10.6 |
| 2 | 8.7 | 9.3 | 2305 | 8.6 | 8.6 | 1.6 | 18.9 | 5.4 | 11.9 |
| 3 | 9.7 | 10.4 | 3394 | 8.4 | 8.2 | 2.5 | 29.6 | 3.4 | 13.4 |
| 4 | 9.5 | 10.2 | 3173 | 9.1 | 9.3 | 2.4 | 26.5 | 4.3 | 13.9 |
| 5 | 4.0 | 4.2 | 2133 | 4.3 | 4.3 | 1.5 | 34.1 | 1.4 | 7.3 |
| 6 | 5.9 | 6.3 | 3787 | 6.4 | 6.5 | 1.7 | 26.2 | 3.1 | 9.8 |
| 7 | 10.1 | 10.8 | 3814 | 10.8 | 10.8 | 1.7 | 15.7 | 7.4 | 14.1 |
| 8 | 12.7 | 13.7 | 3835 | 14.1 | 14.1 | 1.7 | 12.4 | 10.6 | 17.6 |
| 9 | 12.3 | 13.2 | 3855 | 13.0 | 12.9 | 2.2 | 16.6 | 8.7 | 17.3 |
| 10 | 6.4 | 6.9 | 2349 | 8.2 | 8.2 | 1.5 | 18.9 | 5.1 | 11.2 |
| 11 | 4.9 | 5.1 | 2801 | 6.7 | 6.8 | 1.6 | 23.8 | 3.5 | 9.9 |
| 12 | 9.3 | 9.9 | 2568 | 10.5 | 10.7 | 2.2 | 20.7 | 6.2 | 14.9 |

[a] Relative standard deviation (RSD) or coefficient of variation defined as the ratio of the standard deviation (SD) to the mean. [b] The minimum and maximum are defined here as $\mu - 2\sigma$ and $\mu + 2\sigma$, in order to reduce the impact of outliers.

**Table 7.** Tropospheric NO₂ VCD statistics for coincident TROPOMI and APEX pixels for Flight #2, orbit 08826.

| NO₂ VCD$_{TROPO}$ (x 10$^{15}$ molec cm$^{-2}$) | | | NO₂ VCD$_{APEX}$ (x 10$^{15}$ molec cm$^{-2}$) | | | | | | |
|---|---|---|---|---|---|---|---|---|---|
| Pixel ID | VCD$_{TROPO}$ | VCD$_{TROPO-CRE}$ | Count | Mean | Median | SD | RSD(%) | Min | Max |
| 1 | 6.3 | 7.8 | 2082 | 8.1 | 8.0 | 1.8 | 22.6 | 4.4 | 11.6 |
| 2 | 5.6 | 6.8 | 2882 | 7.5 | 7.4 | 1.7 | 22.4 | 4.1 | 10.8 |
| 3 | 4.5 | 5.4 | 2522 | 5.6 | 5.5 | 1.7 | 31.7 | 2.0 | 8.9 |
| 4 | 8.5 | 11.0 | 2843 | 12.6 | 12.8 | 2.7 | 21.3 | 7.3 | 18.3 |
| 5 | 9.7 | 12.4 | 2870 | 12.8 | 12.9 | 2.2 | 17.1 | 8.5 | 17.3 |
| 6 | 9.2 | 11.6 | 2871 | 11.7 | 12.0 | 3.2 | 26.6 | 5.6 | 18.3 |
| 7 | 6.9 | 8.4 | 2882 | 9.3 | 8.4 | 4.3 | 51.4 | -0.2 | 17.0 |
| 8 | 5.1 | 6.1 | 2887 | 5.8 | 5.5 | 1.9 | 33.5 | 1.8 | 9.2 |
| 9 | 3.4 | 4.0 | 2887 | 4.8 | 4.7 | 1.5 | 31.2 | 1.8 | 7.6 |
| 10 | 5.8 | 7.5 | 1882 | 8.7 | 8.7 | 2.2 | 25.6 | 4.2 | 13.1 |
| 11 | 8.0 | 10.2 | 2874 | 8.7 | 8.6 | 1.9 | 22.4 | 4.7 | 12.4 |
| 12 | 6.1 | 7.9 | 2881 | 8.2 | 8.1 | 2.1 | 26.2 | 3.8 | 12.3 |
| 13 | 6.0 | 7.6 | 2888 | 7.1 | 6.9 | 2.2 | 32.1 | 2.5 | 11.3 |
| 14 | 4.1 | 5.0 | 2888 | 5.4 | 5.4 | 1.5 | 28.2 | 2.3 | 8.4 |
| 15 | 4.9 | 6.0 | 2896 | 5.1 | 4.9 | 1.8 | 35.5 | 1.4 | 8.5 |
| 16 | 4.0 | 4.8 | 2637 | 5.1 | 5.0 | 2.0 | 41.0 | 0.9 | 9.0 |
| 17 | 5.7 | 7.4 | 2810 | 6.8 | 6.9 | 1.9 | 28.2 | 3.0 | 10.7 |
| 18 | 5.6 | 7.2 | 2886 | 7.6 | 7.3 | 2.3 | 31.8 | 2.6 | 11.9 |
| 19 | 5.5 | 7.0 | 2771 | 5.9 | 5.6 | 2.4 | 42.6 | 0.8 | 10.4 |
| 20 | 4.1 | 5.1 | 2406 | 5.3 | 5.2 | 2.0 | 38.5 | 1.2 | 9.1 |
| 21 | 4.0 | 4.9 | 1746 | 5.4 | 5.3 | 2.1 | 38.9 | 1.2 | 9.5 |





**Table 8.** Tropospheric $NO_2$ VCD statistics for coincident TROPOMI and APEX pixels for Flight #3, orbit 08840.

| | $NO_2$ $VCD_{TROPO}$ (x $10^{15}$ molec cm$^{-2}$) | | $NO_2$ $VCD_{APEX}$ (x $10^{15}$ molec cm$^{-2}$) | | | | | | |
|---|---|---|---|---|---|---|---|---|---|
| Pixel ID | $VCD_{TROPO}$ | $VCD_{TROPO-CRE}$ | Count | Mean | Median | SD | RSD(%) | Min | Max |
| 1 | 4.2 | 4.7 | 1562 | 4.7 | 4.6 | 2.3 | 47.6 | 0.2 | 9.2 |
| 2 | 7.2 | 8.2 | 1929 | 8.6 | 8.6 | 2.1 | 24.0 | 4.5 | 12.7 |
| 3 | 6.8 | 7.7 | 2693 | 7.2 | 7.2 | 1.9 | 26.7 | 3.4 | 11.1 |
| 4 | 5.2 | 5.9 | 2912 | 6.3 | 6.3 | 1.7 | 26.6 | 3.0 | 9.7 |
| 5 | 4.0 | 4.4 | 2898 | 6.0 | 6.0 | 1.8 | 29.8 | 2.4 | 9.6 |
| 6 | 4.8 | 5.2 | 2511 | 5.5 | 5.5 | 1.7 | 31.9 | 2.0 | 8.9 |
| 7 | 8.3 | 9.5 | 2870 | 9.2 | 9.2 | 2.0 | 21.3 | 5.3 | 13.1 |
| 8 | 8.4 | 9.6 | 2926 | 8.7 | 8.6 | 2.0 | 22.6 | 4.8 | 12.6 |
| 9 | 7.5 | 8.5 | 2919 | 8.3 | 8.4 | 1.9 | 22.2 | 4.6 | 12.0 |
| 10 | 6.9 | 7.7 | 2910 | 7.9 | 7.9 | 1.6 | 20.9 | 4.6 | 11.2 |
| 11 | 6.5 | 7.2 | 2907 | 7.5 | 7.5 | 1.9 | 26.0 | 3.6 | 11.4 |
| 12 | 4.9 | 5.3 | 2792 | 5.7 | 5.6 | 1.7 | 30.4 | 2.2 | 9.2 |
| 13 | 5.6 | 6.4 | 2290 | 5.1 | 5.0 | 1.8 | 35.7 | 1.5 | 8.7 |
| 14 | 5.0 | 5.7 | 1668 | 5.2 | 5.2 | 1.8 | 35.0 | 1.6 | 8.9 |
| 15 | 5.1 | 5.6 | 2018 | 5.2 | 5.2 | 1.5 | 28.7 | 2.2 | 8.2 |

**Table 9.** Tropospheric $NO_2$ VCD statistics for coincident TROPOMI and APEX pixels for Flight #4, orbit 08854.

| | $NO_2$ $VCD_{TROPO}$ (x $10^{15}$ molec cm$^{-2}$) | | $NO_2$ $VCD_{APEX}$ (x $10^{15}$ molec cm$^{-2}$) | | | | | | |
|---|---|---|---|---|---|---|---|---|---|
| Pixel ID | $VCD_{TROPO}$ | $VCD_{TROPO-CRE}$ | Count | Mean | Median | SD | RSD(%) | Min | Max |
| 1 | 4.7 | 5.5 | 2378 | 6.9 | 6.8 | 2.1 | 30.7 | 2.7 | 11.2 |
| 2 | 4.5 | 5.3 | 3786 | 5.0 | 4.8 | 1.9 | 37.7 | 1.2 | 8.7 |
| 3 | 7.3 | 8.5 | 3871 | 7.2 | 5.9 | 5.0 | 69.0 | 1.0 | 17.2 |
| 4 | 6.3 | 7.2 | 3230 | 7.6 | 6.8 | 3.5 | 45.8 | 0.6 | 14.5 |
| 5 | 10.0 | 11.6 | 3998 | 9.3 | 8.5 | 3.8 | 41.0 | 1.7 | 16.9 |
| 6 | 14.5 | 16.9 | 3973 | 17.1 | 17.5 | 5.0 | 29.5 | 7.0 | 27.2 |
| 7 | 6.6 | 7.5 | 3686 | 8.1 | 6.7 | 4.6 | 57.1 | 1.0 | 17.3 |
| 8 | 11.2 | 12.8 | 3194 | 15.0 | 15.2 | 3.3 | 22.3 | 8.3 | 21.7 |
| 9 | 11.7 | 13.3 | 3976 | 17.6 | 17.6 | 3.1 | 17.4 | 11.5 | 23.7 |
| 10 | 5.5 | 6.1 | 3418 | 6.3 | 5.3 | 3.7 | 59.0 | 1.0 | 13.8 |

