# Peer review of "Assessment of the TROPOMI tropospheric NO2 product based on airborne APEX observations"

_Atmospheric Measurement Techniques, 2020_

## Short Comment (SC1) · 30 Jun 2020

P3:30 Kim et al.(2017) was cited as a reference for GEMS. More appropriate reference for GEMS (and balanced with TEMPO's) is Kim et al.(2020),

Kim, J., and Coauthors: New Era of Air Quality Monitoring from Space: Geostationary Environment Monitoring Spectrometer (GEMS). Bull. Amer. Meteor. Soc., 101, E1–E22, https://doi.org/10.1175/BAMS-D-18-0013.1, 2020.
* * *

---

## Short Comment (SC2) · 1 Jul 2020

Thank you for the comment. We will take this into account for the revised version of the paper

---

## Referee Comment (RC1) · Anonymous Referee #1 · 27 Jul 2020

General Comments about the manuscript in discussion titled, 'Assessment of the TROPOMI tropospheric NO2 product based on airborne APEX observations' by Tack et al., 2020. This manuscript is a well thought out analysis which uses the airborne instrument, APEX, to evaluate tropospheric NO2 columns from TROPOMI over two major Belgian cities, Antwerp and Brussels. This work does a great job in assessing the impact of spatial resolution between the observations and a lot of detailed analysis on spatial smearing and found that the NO2 tropospheric NO2 product is within the precision and accuracy requirements. Overall, the analysis is quite detailed and very fitting for publication within AMT. I recommend publishing after the proper addressing a couple specific concerns pertaining to details about the APEX retrieval and some other

minor comments.

Specific Comments: There is concern about the assumption of vertical NO2 profile in the APEX retrieval as well mixed profile of NO2 through the boundary layer. There have been many observations and analysis in the literature proving that NO2 is rarely 'well-mixed' in an urban environment (e.g., http://dx.doi.org/10.1002/2015JD024203). (1) There appears to be modeled high resolution model data available from the regional CAMS model that likely at last has some more realistic weighting of NO2 nearer to the surface (negative vertical gradient in the boundary layer). The analysis would be strengthened if results were also shown with those a priori in the APEX retrieval. (2)Alternatively or in addition, the analysis would also be strengthened if there was some background on the validation of APEX NO2 observations or perhaps independent validation with measurements from the MAX-DOAS measurements mentioned in this analysis. It is hard to evaluate TROPOMI bias if the reference measurement is not validated itself.

There are some missing details about the APEX NO2 tropospheric column algorithm. Please add discussion about the reference spectra (i.e., is there one per flight? One overall? Where is it? I saw the comment that it was estimated using a mobile MAX-DOAS) also please add some text that discusses how APEX tropospheric vertical columns are computed (e.g., is it similar to Sect. 3.2.2 and 3.3 in Lamsal et al. http://dx.doi.org/10.1002/2016JD025483 ?)

How is sigmaAMF_APEX computed?

It is interesting in Table 4 how the bias/slopes are different between the two cities. Antwerp has a lower slope for all three column comparisons as well as a larger negative bias. Any comment on this?

On page 5, there is discussion about AOT measurements. Were any observed in Antwerp or only in Brussels?

In Figure 15 and Sect. 6.2: why does the color bar go to zero if the background is 3x10ˆ15 and the detection limit is assumed at 5.1x10ˆ15? I am not sure if this is an oversight or if the section needs some clarifying discussion about the interpretation of this figure.

Page 1 Line 31 and generally in the paper: These biases are for these Belgian cities but are stated as general results for 'urban areas'. Could these results perhaps be different in other cities?

Technical Comments: Page 1: Line 23: You refer to the slope of 0.93 after the introduction of the CAMS profile, however the original slope is not listed. Please add this to the abstract to be consistent.

Page 3 Line 1: please add the TROPOMI resolution sooner than is mentioned in page 3 line 15 as it is referenced in relation to other missions.

Page 3: Please consider swapping the placement of the second and third paragraphs in this page (Paragraph 2 being 'In this study...' and Paragraph 3 being 'Richter et al...' ). It would improve flow as it talks about the challenges then state how this study addresses those challenges

Page 3 Line 31:There is this reference also in AMTD. https://amt.copernicus.org/preprints/amt-2020-151/ Perhaps make the statement more defining to the region studied or other details. Or remove/edit accordingly.

Page 5 Line 30: AURA should be Aura. It is not an acronym. Same with PANDORA–>Pandora.

Page 7 Line 13-14: 'is based' is used twice in one sentence.

Page 9 Final paragraph: This figure shows the difference in Box AMFs based on albedo, and therefore belongs better in the next section rather than Sect. 4.3.1 about A priori NO2 profiles.

[Figure]

Page 12: Line 20: Word Choice: refer to Antwerp and Brussels as regions or cities, rather than separate campaigns.

Figure 7: please point out the airport for ease of identifying when discussed in the text on Page 13

Page 14 Line 21-22: It is premature to make a statement about the error bars in Figure 8 since the figure is not introduced until a couple pages later. I suggest removing that sentence here.

Page 18 Lines 15-23: Please clarify this discussion on how the temporal variability between TROPOMI overpasses is computed, especially with the differences in pixel footprints. It is hard to follow what those statistics are referring to and how they are computed.

Page 19 Line 10: delete 'allow to'

Figure 1: Adding a label for Stabroek as the other ground site where meteorology is measured in Antwerp could be helpful.

Figure 13: Please make the red dots more visible. (Perhaps white like in other Figures). Also in the caption write what they are. And as a suggestion, pull the color bar legend out of panel (a) and make larger since it refers to all four maps.

[Figure]

---

## Referee Comment (RC2) · Anonymous Referee #2 · 28 Jul 2020

The manuscript by Tack et al. "Assessment of the TROPOMI tropospheric NO2 product based on airborne APEX observations", is well-written with a solid methodology. The accommodating figures are of excellent quality and easy to understand. The study assesses the quality of the TROPOMI NO2 observations and compares them to high-resolution aircraft-borne remote sensing observations over two Belgian cities. The impact of spatial smoothing and NO2 a priori is estimated and discussed in great detail. Overall, I think this is a great fit for the TROPOMI special issue in AMT and I would recommend publication. I have some minor comments and suggestions that should be addressed.

[Figure]

General comments

p.3 l. 31: Is this really the first? Around the same time: Evaluating Sentinel-5P TROPOMI tropospheric NO2 column densities with airborne and Pandora spectrometers near New York City and Long Island Sound, Laura M. Judd et al.; https://doi.org/10.5194/amt-2020-151; I would suggest deleting this comment and possibly include a reference to this paper within the manuscript.

p. 4 l. 16: The TROPOMI tropospheric columns are up to ~12km. There is NO2 above 6.5km, the NO2 profile is not 0. Over cities and enhanced areas this will not be a big factor, but this should be discussed and mentioned. A typical amount of NO2 from 6.5 to 12km over Belgium would be useful to mention – maybe using CAMS or TMP.

p. 4/5 I think the cloud fraction should be mentioned. It's mentioned that for the flights it was mainly clear sky, but what is the range of the cloud fractions for the TROPOMI observations? Some of this could be of course due to aerosols, but I think it would be good to know the cloud fraction (nitrogen dioxide window) assumed in the TROPOMI retrieval. I just noticed this is mentioned later in Sect. 4.3.3 , but it would be good to include it in this section.

p.10 l. 1 Could the difference of the AMF come from the different height? APEX is from the surface to 6.5km; for TROPOMI it's higher.

Sect 4.3.2 Albedo: the TROPOMI AMF could be re-calculated using the APEX albedo and the impact can be directly estimated. I think the study would benefit from looking at the impact of correcting for the albedo. I'm not sure if this would be possible to do within a reasonable amount of time, if this is too time consuming, just mention it at least.

Sect. 4.3.2: The albedo is wavelength dependent; albedos at 3 different wavelengths are compared. How big is the impact of the wavelengths difference? This should be discussed, e.g. look at the OMI albedo and include the relative difference for these

different wavelengths (over Brussels and Antwerp).

p. 13, l. 16: it could be due to meteorology; e.g. lower wind speeds can increase the VCD enhancement even though emissions do not increase VCDs can be higher for stagnant winds, there could also be factors that potentially increased the lifetime of NOx (e.g. OH, O3, and NOx concentration) for that particular day. I think meteorology should be mentioned as a potential influence; look at the wind speeds and direction for these days (the wind speed is definitely lower). If both TROPOMI and APEX observed higher VCDs on June 29, this would not be due to the APEX instrument troubles.

p. 22, l. 21: Can you really conclude this if your comparison is done over a small area and over a short time period, seasons are not considered (e.g. snow)? Re-phrase this, or add "over Belgium in the summer time."

Specific/technical comments

p.1 l. 24-26: "When the absolute value …, when comparing APEX NO2 VCDs with TM5-MP based and CAMS-based NO2 VCDs, respectively." I suggest re-wording this sentence, e.g.: The absolute difference is on average xx molec cm-2 (16%) and xx molec cm-2 (9%) compared to …

p.1 l. 26: Which accuracy requirement; maybe change it to "mission accuracy requirement"

p.1 l. 29-30; suggest re-wording: Something like: The current TROPOMI data underestimate localized enhancements and overestimate background values by approximately 1-2x 1015 molec cm-2 (10- 20%).

p.3 l. 13: "studied in Sect. 6" change to "see Sect. 6"

p.4 l. 4 Air pollution levels over Belgium… Do you have a reference that can be included here?

p.6, l. 16 VNIR; this should be defined, maybe in the previous sentence were the two

channels are mentioned

p.7 l. 28, mention the height of the layers (between surface and xx km)

---

## Author Comment (AC1) · 7 Oct 2020

**Anonymous Referee #1:**

We greatly appreciate the positive feedback from the referee and the constructive comments. As described below, we have modified the manuscript according to suggestions and clarified where necessary. We hope that the revised manuscript has improved in respect to the original paper. Please find a rebuttal against each point below.

***Black, bold, italic: Referee's comments***

Black: Author's reply

Changes in the original discussion paper are highlighted in yellow and attached below

***1) There is concern about the assumption of vertical NO2 profile in the APEX retrieval as well mixed profile of NO2 through the boundary layer. There have been many observations and analysis in the literature proving that NO2 is rarely 'well-mixed' in an urban environment (e.g., http://dx.doi.org/10.1002/2015JD024203). (1) There appears to be modeled high resolution model data available from the regional CAMS model that likely at last has some more realistic weighting of NO2 nearer to the surface (negative vertical gradient in the boundary layer). The analysis would be strengthened if results were also shown with those a priori in the APEX retrieval.***

It is true that in most studies assumptions are made on the profile shape in the boundary layer as high resolution model profiles are not always available (and also can have significant errors). Most campaigns involving airborne spectrometers are also lacking measurements of the vertical gas distribution as it requires an additional set of in-situ instruments and specific flight patterns. Note that we are involved in a project to address/study this problem by combining a spectrometer and in-situ instruments in one aircraft (RAMOS - http://environment.inoe.ro/article/179/about-ramos). The aircraft will also execute flights over Bucharest, Romania in 2020-2021 in the context of TROPOMI validation. This data set will allow us to better assess the impact of measured, modelled or assumed well-mixed profiles.

As indicated in Sect. 4.3.1, the decision was taken to use box profiles for the reference APEX retrievals in order to be independent from both the standard TROPOMI product based on TM5-MP profiles, and the TROPOMI product based on CAMS profiles. In the paper a sensitivity study was already included where the box profiles were replaced by interpolated TM5-MP profiles. We have followed your suggestion and also assessed the impact of replacing the box profiles by CAMS profiles. The findings are in line with a previous study (Tack et al. 2017) where we assessed the impact on the APEX retrievals of using high resolution a priori $NO_2$ profiles from the 1 km x 1 km AURORA model instead of box profiles.

In Sect. 4.3.1, we have changed the paragraph accordingly:
"For the APEX retrievals, AEPs and a priori $NO_2$ profiles were constructed from the AOT and PBL height observations, as discussed in Sect. 4.2. In order to yield retrievals independent from the satellite, box profiles were used instead of the TROPOMI TM5-MP profiles, as displayed in Fig. 3a. When TM5-MP or CAMS profiles would be applied as a priori for the APEX retrievals, the AMF would increase with respectively 9% and 10% on average, which is largely consistent with a similar sensitivity study reported in Tack et al. (2017). For the APEX retrievals, we assumed a well-mixed $NO_2$

and aerosol box profile scenario and urban aerosols with a high single-scattering albedo (SSA) of 0.93. This causes a multiple scattering scenario and an enhancement of the optical path length in the $NO_2$ layer, and results in an increase in the AMF. When instead considering a no aerosol scenario for the APEX retrievals, the AMF drops by 10% on average. We assume that the opposing effects of using (1) a priori profile shape assumptions different from the TROPOMI retrievals and (2) different aerosol assumptions tend to cancel each other out in the APEX retrievals."

*2) Alternatively or in addition, the analysis would also be strengthened if there was some background on the validation of APEX NO2 observations or perhaps independent validation with measurements from the MAX-DOAS measurements mentioned in this analysis. It is hard to evaluate TROPOMI bias if the reference measurement is not validated itself.*

Validation implies that the reference data has a better accuracy than the data set to be validated. This is indeed the case for the MAX-DOAS data when compared to the airborne APEX data. However, there is the issue of differences in horizontal representativity and potential sampling of different air masses.

For the overpasses over the MAX-DOAS station on 26 and 28 June we have compared the MAX-DOAS and APEX retrievals. We have only two overpasses in this data set, but we hope to include more (MAX-)DOAS instruments during the follow-up campaign in summer 2021.

Note as well that APEX $NO_2$ VCD retrievals have been assessed and validated by comparison with other airborne imagers, as well as GB DOAS measurements during the AROMAPEX intercomparison campaign reported in https://doi.org/10.5194/amt-12-211-2019. This is mentioned in the introduction of the study under review.

We have added a discussion on the comparison with MAX-DOAS at the end of Sect. 5.2.2:
"For the flights over the Brussels region, we have also compared the TROPOMI and APEX $NO_2$ VCD with the MAX-DOAS $NO_2$ VCD at the time of overpass and results are provided in Table 5. The TROPOMI $NO_2$ VCD is provided for the pixel in which the station resides for both the TM5-MP-based and CAMS-based product. The APEX $NO_2$ VCD is provided for the average within the TROPOMI pixel footprint over the MAX-DOAS station and for the specific APEX pixel over the station. As the MAX-DOAS is performing elevation scans in a fixed azimuth direction (35° N), APEX observations are also averaged along this line of sight (LOS) in order to take into account the instrument directivity and in order to reduce potential mismatches due to differences in spatial representativity. In this case, however, temporal mismatches can occur as APEX pixels, acquired in different flight lines, are averaged. Based on the study of Dimitropoulou et al. (2020), the horizontal sensitivity of the MAX-DOAS is estimated to be in the order of 10 km for measurements in Brussels in summer time and in the visible wavelength range. MAX-DOAS observations are filtered based on the degrees of freedom (DOFs) which should be larger than two. Secondly, the relative root mean square error (RMSE) of the difference between measured and calculated differential slant column densities with respect to the zenith spectrum of each scan should be smaller than 15 % (Dimitropoulou et al., 2020). On 26 June there is clearly a pollution event not seen over the station but further northeast along the MAX-DOAS LOS, as can be observed in the APEX $NO_2$ VCD grid (see Fig. 7a and Fig. 11). When averaging the APEX pixels along the MAX-DOAS LOS, the difference in MAX-DOAS and APEX $NO_2$ VCD is reduced from 4.8 to 0.1 x $10^{15}$ molec $cm^{-2}$. On June 28, the diurnal variation in the $NO_2$ field is much smaller. We see a slight underestimation of 0.3 x $10^{15}$ molec $cm^{-2}$ for the APEX observation above the station

when compared to MAX-DOAS, while the latter is overestimated by 1.2 x 10$^{15}$ molec cm$^{-2}$ when averaging along the LOS."

**Table 5.** Co-located TROPOMI, APEX and MAX-DOAS observations for the flights over Brussels. The TROPOMI NO$_2$ VCD is provided for the pixel in which the MAX-DOAS station resides for both the TM5-MP-based and CAMS-based product. The APEX NO$_2$ VCD is provided for the average within the TROPOMI pixel footprint over the MAX-DOAS station and for the specific APEX pixel over the station. As the MAX-DOAS is performing elevation scans in a fixed azimuth direction (35° N), APEX observations are also averaged along this line of sight in order to take into account the instrument directivity.

| | Flight #1 (26-06-2019) | | Flight #3 (28-06-2019) | |
|---|---|---|---|---|
| NO$_2$ VCD$_{TROPO}$ pixel over MAX-DOAS station [a] (x 10$^{15}$ molec cm$^{-2}$) | 8.7 | | 6.8 | |
| NO$_2$ VCD$_{TROPO}$-CRE pixel over station [a] (x 10$^{15}$ molec cm$^{-2}$) | 9.3 | | 7.7 | |
| NO$_2$ VCD$_{APEX}$ (x 10$^{15}$ molec cm$^{-2}$) | | | | |
|     Averaged in TROPOMI pixel over station | 8.6 | | 7.2 | |
|     APEX pixel over station | 8.4 | | 6.4 | |
|     APEX pixels averaged along MAX-DOAS viewing direction | 13.1 | | 7.9 | |
| | TROPOMI overpass (14:56 LT) | APEX overpass (14:07 LT) | TROPOMI overpass (14:19 LT) | APEX overpass (14:25 LT) |
| NO$_2$ VCD$_{MAX-DOAS}$ (x 10$^{15}$ molec cm$^{-2}$) | 25.0 | 13.2 | 6.7 | 6.7 |

[a] TROPOMI Pixel ID #2 in Table 7 for Flight #1 and Pixel ID #3 in Table 9 for Flight #3.

***3) There are some missing details about the APEX NO2 tropospheric column algorithm. Please add discussion about the reference spectra (i.e., is there one per flight? One overall? Where is it? I saw the comment that it was estimated using a mobile MAXDOAS) also please add some text that discusses how APEX tropospheric vertical columns are computed (e.g., is it similar to Sect. 3.2.2 and 3.3 in Lamsal et al. http://dx.doi.org/10.1002/2016JD025483 ?)***

APEX NO$_2$ VCD retrievals are deliberately not discussed in full detail here as this has been done extensively in Tack et al. (2017) (https://doi.org/10.5194/amt-10-1665-2017) and also partly in Tack et al. (2019) (https://doi.org/10.5194/amt-12-211-2019). Tack et al. (2017) focuses on the development of the APEX NO$_2$ retrieval algorithm (which is indeed similar in concept to Lamsal et al. (2017)) and is applied on data acquired in 2015 over the Antwerp and Brussels region. The developed retrieval algorithm has been applied to the data acquired for the study under review. We prefer to avoid repetition and a too lengthy paper and want to keep the focus on the actual comparison/validation and study on impact of spatial resolution. Having a full discussion again on the APEX retrieval would be out of scope for this paper and it would similarly require a full discussion on the TROPOMI retrievals. We assume that the retrieval algorithms are well documented for both, TROPOMI retrievals in the ATBD and APEX retrievals in Tack et al. (2017). We have adapted Sect. 4.1 and 4.2 in such a way to emphasize why we don't include a full discussion on the retrieval algorithm and highlighted explicit references to the relevant sections in Tack et al. (2017) for the readers, interested in more details about the APEX retrievals.

For each flight, a reference spectrum was selected in a clean background area, upwind of the main sources, and the residual amount of NO$_2$ in the reference was estimated from co-located mobile-DOAS measurements. This has also been added to Sect. 4.2.

We have updated Sect. 4.2 as follows:

"The APEX NO$_2$ VCD retrieval scheme is similar in concept to the TROPOMI one and the developed algorithm is well documented in Tack et al. (2017). A full discussion on the retrieval algorithm is beyond the scope of this paper. Therefore, we refer to Sect. 4.1, Sect. 4.2, Sect. 4.3, and Sect. 4.6 in Tack et al. (2017) for all details on the APEX DOAS analysis, reference spectrum, AMF computation, and NO$_2$ VCD error budget, respectively. The DOAS spectral fit is based on the QDOAS software (Fayt et al., 2016) applied in the 470-510 nm spectral range, optimal for NO$_2$ retrieval from APEX. Note that interference with unidentified instrumental artefacts or features prevents us from extending the fitting window to wavelengths lower than 470 nm as discussed in Popp et al. (2012) and Tack et al. (2017). Key parameters for the NO$_2$ SCD retrieval are provided in Table 3. For each flight, a reference spectrum was selected in a clean background area, upwind of the main sources, and the residual amount of NO$_2$ in the reference was estimated from co-located mobile-DOAS measurements. …"

**4) How is sigmaAMF_APEX computed?**

Similarly as for comment 3, the APEX NO$_2$ VCD uncertainty budget is not discussed in full detail here as this has been done extensively in Tack et al. (2017) (https://doi.org/10.5194/amt-10-1665-2017) and also partly in Tack et al. (2019) (https://doi.org/10.5194/amt-12-211-2019). However, we agree more details should be added here, as well as clear references for readers that would like to have a full discussion.

We would like to refer to Section 4.6 in Tack et al. (2017) (https://doi.org/10.5194/amt-10-1665-2017):

"The error in the calculation of the air mass factor σAMF$_i$ is caused by the uncertainties in the assumptions made for the radiative transfer model parameters (See Sect. 4.3.1). The contributing uncertainties can be summed in quadrature to obtain an overall error estimate σAMFi . According to Boersma et al. (2004), the error budget associated with the computation of the AMF is dominated by the cloud fraction, surface albedo and NO2 profile shape: (1) as flights took place under clear-sky conditions, cloud fraction is not considered an error source in this case. (2) Sensitivity tests, performed in Sect. 4.3.2, indicate that the surface albedo has the most significant impact on the effective light path, thus on the AMF. Within the albedo 1σ interval, the AMF variability can be up to 65 %. However, as absolute radiances can be directly derived from the APEX instrument, the albedo can be determined with relatively high accuracy. For a realistic estimate of the uncertainty, the following study was performed: several albedo types were measured in the field with an ASD FieldSpec-4 spectrometer (http://www.asdi.com/products-and-services/ fieldspec-spectroradiometers/fieldspec-4-hi-res) and compared to the APEX surface albedo. For the wavelength 490 nm, the average albedo error over all targets is 10 %, which is assumed to be a realistic estimate of the uncertainty related to the a priori surface albedo. (3) Based on the sensitivity study performed in Sect. 4.3.2, the uncertainty related to the a priori NO2 profile shape is lower than 8 %. (4) According to the performed simulations, the uncertainty related to the assumption of a pure Rayleigh atmosphere is estimated to be less than 10 %. (5) Both the viewing and sun geometry can be determined with high accuracy, thus the impact on the error in the AMF computation is expected to be small. Moreover, the performed sensitivity study, summarised in Table 5, has revealed that varying input for the viewing/sun geometry has a very low impact on the TAMF variability. Therefore it is assumed that the uncertainties related to RAA, VZA and SZA are less than 1 %. Finally, all error sources contributing to the overall error σAMFi are summed in quadrature and an estimate of approximately 15 % is obtained."

We have added more details on this in the manuscript as follows: "A full error budget for APEX NO$_2$ VCD retrievals has been discussed in Sect. 4.6 in Tack et al. (2017). Like for TROPOMI, the overall error on the retrieved APEX NO$_2$ VCDs, σVCD$_{APEX}$, is dominated by uncertainties related to the DOAS fit and AMF computation. The error on the retrieved DSCD or the slant error, σDSCD$_{APEX}$, estimated

from the fit residuals in the DOAS analysis, is 3.1 x $10^{15}$ molec cm$^{-2}$, on average. The error on the AMF computation, σAMF$_{APEX}$, depends on uncertainties in the assumption of the RTM inputs with respect to the true atmospheric state and is dominated by systematic errors in the surface albedo, NO$_2$ profile, and aerosol parameters. An estimate of approximately 15% is obtained for σAMF$_{APEX}$, following the detailed error budget described in Sect. 4.6 in Tack et al. (2017)."

*5) It is interesting in Table 4 how the bias/slopes are different between the two cities. Antwerp has a lower slope for all three column comparisons as well as a larger negative bias. Any comment on this?*

Your observation is correct. We checked the individual correlation plots for the different flights and it is hard to give a conclusive explanation based on the current data sets. The main difference between the two data sets is the type of emissions: prevailing industrial emissions in Antwerp and more traffic emissions in Brussels. This leads to a larger dynamic range and heterogeneity in the NO$_2$ field for the Antwerp region. Even if the APEX measurements are averaged within the TROPOMI pixel footprints, this still might have an effect for example due to the non-perfect time coincidence, point spread function, local albedo variability, etc. However, note that the correlation coefficient does not seem to be affected. It is hard to say as we don't have enough statistics. As new flights over both areas are expected in summer 2021, we hope to be able to check this again if it is a coincidence or really something geophysical.

*6) On page 5, there is discussion about AOT measurements. Were any observed in Antwerp or only in Brussels?*

Unfortunately no AOT measurements were done in Antwerp. Due to restricted national funding, this was a "lightweight" campaign and we relied on existing ground-based stations like the CIMEL and MAX-DOAS station we have in Uccle. A new S5P validation is scheduled in summer 2021 based on ESA funding which would give is more room to invite other teams and maybe add additional instruments in the two regions.

*7) In Figure 15 and Sect. 6.2: why does the color bar go to zero if the background is 3x10ˆ15 and the detection limit is assumed at 5.1x10ˆ15? I am not sure if this is an oversight or if the section needs some clarifying discussion about the interpretation of this figure.*

Thanks for pointing this out. We took indeed a standard color bar between 0 and 5 x 10ˆ16 molec cm$^{-2}$ while the data shown is only ranging between 0.51 and 5 x 10ˆ16. Synthetic NO$_2$ VCDs below the detection limit of 5.1 x 10ˆ15 molec cm$^{-2}$ are masked white and indeed even without masking, the lowest values would be 0.3 x 10ˆ16 molec cm$^{-2}$ and not 0. But note that no VCD values in the plot had the deep blue colors representing 0 to 0.51 x 10ˆ16 molec cm$^{-2}$. To avoid any confusion we have adapted the colorbar with limits between 0.5(1) and 5 x 10ˆ16.

*8) Page 1 Line 31 and generally in the paper: These biases are for these Belgian cities but are stated as general results for 'urban areas'. Could these results perhaps be different in other cities?*

Indeed, this can be certainly different for other cases, depending on the amount of heterogeneity in the NO$_2$ field as well as the satellite pixel size (at nadir or more at edge of the swath). These nuances are well discussed in Sect. 6.1, also with reference to other studies. But indeed the statement in the abstract is "too strong" like this. We have adapted this in the abstract to (also following comment #11 from reviewer #2): "For a case study in the Antwerp region, the current TROPOMI data underestimates localised enhancements and overestimates background values by approximately 1-2 x $10^{15}$ molec cm$^{-2}$ (10- 20%)."

For the same reason the related paragraph in the conclusion was adapted to:" The TROPOMI spatial resolution is limited to resolve fine-scale urban $NO_2$ plumes and can cause a considerable smoothing effect in case of the observation of strongly polluted scenes with steep gradients. This depends both on the instrument pixel size and the amount of heterogeneity in the $NO_2$ field. The high-resolution APEX retrievals allow to monitor the effective horizontal variability in the $NO_2$ field at much finer scale. In Sect. 6, the impact of smearing of the effective signal due to the finite satellite pixel size was studied for the Antwerp region based on a downsampling approach of the APEX retrievals. Assuming a pixel size of 25 to 20 $km^2$, equivalent to the initial 3.5 km x 7 km and new TROPOMI 3.5 km x 5.5 km spatial resolution (at nadir), the TROPOMI data underestimates localised enhancements and overestimates urban background values by approximately 1-2 x $10^{15}$ molec $cm^{-2}$, on average, or 10% - 20%, for the Antwerp case study. The average under- and overestimation is further reduced to 0.6-0.9 x $10^{15}$ molec $cm^{-2}$, or smaller than 10%, when increasing the pixel size to 1 $km^2$. Therefore, detailed air quality studies at the city scale still require observations at higher spatial resolution, in the order of 1 $km^2$ or better, in order to resolve all fine-scale structures within the typical heterogeneous $NO_2$ field."

Please see also a related comment (comment #8) from reviewer #2.

**9) Technical Comments: Page 1: Line 23: You refer to the slope of 0.93 after the introduction of the CAMS profile, however the original slope is not listed. Please add this to the abstract to be consistent.**

We suggest to change to "When replacing the coarse 1° x 1° TM5-MP a priori $NO_2$ profiles by $NO_2$ profile shapes from the CAMS regional CTM ensemble at 0.1° x 0.1°, R is 0.94 and the slope increases from 0.82 to 0.93. The bias is reduced to -0.1 ± 1.0 x $10^{15}$ molec $cm^{-2}$ or -1.0% ± 12%."

**10) Page 3 Line 1: please add the TROPOMI resolution sooner than is mentioned in page 3 line 15 as it is referenced in relation to other missions.**

You are right the resolution should be given here. We have moved the sentence from line 15 (initially 3.5 km x 7 km at nadir observations and 3.5 km x 5.5 km since 6 August 2019) and changed the sentence at line 15 to "The APEX spatial resolution is considerably higher than the typical resolution of spaceborne sensors. For example, one TROPOMI pixel of 3.5 km by 7 km comprises approximately 4000 APEX pixels."

**11) Page 3: Please consider swapping the placement of the second and third paragraphs in this page (Paragraph 2 being 'In this study. . .' and Paragraph 3 being 'Richter et al. . .' ). It would improve flow as it talks about the challenges then state how this study addresses those challenges**

Thank you for the suggestion. We agree swapping the two paragraphs improves the flow.

**12) Page 3 Line 31: There is this reference also in AMTD. https://amt.copernicus.org/preprints/amt-2020-151/ Perhaps make the statement more defining to the region studied or other details. Or remove/edit accordingly.**

The study https://doi.org/10.5194/amt-2020-**151** was indeed submitted to AMT in the same week as the study under review (https://doi.org/10.5194/amt-2020-**148**). We have adapted the paragraph in the manuscript and we have added a proper reference, now it is available:

This is one of the first publications assessing TROPOMI $NO_2$ retrievals over strongly polluted regions based on the comparison with airborne remote sensing observations and it is one of the first airborne spectrometer data sets coinciding in space and time with a large amount of fully sampled satellite pixels. At the same time the study of Judd et al. (2020) on the Long Island Sound Tropospheric Ozone Study (LISTOS) campaign in the New York City/Long Island Sound region has been submitted. Earlier studies reporting on the validation of spaceborne observations based on airborne spectrometer data, such as Heue et al. (2005), Constantin et al. (2016), Lamsal et al. (2017), Broccardo et al. (2018), and Merlaud et al. (2020) have shown high potential but are scarce, mainly due to the relatively large pixel footprint of TROPOMI's predecessors with respect to the area that can be covered with an airborne mapping spectrometer.

*13) Page 5 Line 30: AURA should be Aura. It is not an acronym. Same with PANDORA– >Pandora.*

Thanks for clearing this out. This is corrected throughout the manuscript.

*14) Page 7 Line 13-14: 'is based' is used twice in one sentence.*

Corrected to:

"The processor is based on a retrieval-data assimilation-modelling system using the 3-D global TM5-MP chemistry transport model (CTM) (Williams et al., 2017). It follows a 3-step approach: "

*15) Page 9 Final paragraph: This figure shows the difference in Box AMFs based on albedo, and therefore belongs better in the next section rather than Sect. 4.3.1 about A priori NO2 profiles.*

We prefer to keep the discussion on the box AMFs (and Figure 3.b) in section 4.3.1 on the $NO_2$ vertical profiles. They are related as Figure 3.a provides the concentration at each altitude layer while the Box AMF in 3.b provides the vertical sensitivity to $NO_2$. It is true that we provide the box AMF profiles for two different albedo scenarios, but the key discussion is on the vertical sensitivity. To make this more clear we suggest to change the title of Sect. 4.3.1 from "A priori $NO_2$ profile" to "$NO_2$ profile and vertical sensitivity".

*16) Page 12: Line 20: Word Choice: refer to Antwerp and Brussels as regions or cities, rather than separate campaigns.*

Indeed referring to it as separate campaigns is not appropriate. We suggest to refer to it as regions here

*17) Figure 7: please point out the airport for ease of identifying when discussed in the text on Page 13*

We have added a white square in Fig. 7 a) and b) and properly referred to it in the caption and text.

*18) Page 14 Line 21-22: It is premature to make a statement about the error bars in Figure 8 since the figure is not introduced until a couple pages later. I suggest removing that sentence here.*

True, we have removed the sentence in this section. Note that in the next section (Sect. 5.2.2), we added an explicit reference to Eq. 1 and 2: "Vertical error bars indicate the overall error in $NO_2$

VCD$_{TROPO}$ (Eq. 1), while the horizontal whiskers represent the error in NO$_2$ VCD$_{APEX}$ retrievals (Eq. 2), averaged over all APEX pixels coinciding with a particular TROPOMI pixel."

**19) Page 18 Lines 15-23: Please clarify this discussion on how the temporal variability between TROPOMI overpasses is computed, especially with the differences in pixel footprints. It is hard to follow what those statistics are referring to and how they are computed.**

Indeed some details for the comparison were missing here. Prior to the comparison we have regridded the data sets to a common grid of 0.1°. In a next step we compared the absolute and relative differences between the two overpasses (grids) on the same day for the full Belgian domain. So the statistics are the average for all "difference pixels" over Belgium. We have clarified this section as follows: " Both on 26 June and 29 June 2019, there were two early-afternoon S-5P overpasses over Belgium with a time difference between the two orbits of approximately 100 min. To assess the impact of the temporal NO$_2$ variability, the changes in the NO$_2$ field have been studied in the subsequent overpasses for the Belgian domain. Prior to the comparison, the data sets have been regridded to a common grid of size 0.1°. On June 26, the absolute value of the differences observed over the full Belgian domain is 3.8 ± 5.3 x 10$^{14}$ molec cm$^{-2}$ or 12% ± 10%, on average. A maximum difference of 5.8 x 10$^{15}$ molec cm$^{-2}$ or 57% was observed for a pixel over the harbor of Antwerp, most likely due to a combination of moving air masses in the key plumes and slight changes in the wind pattern. Additionally, the TROPOMI pixel footprints have different sizes and orientations which also has an effect when sampling the effective NO$_2$ patterns and when regridding to the common grid size of 0.1°. On June 29, the absolute value of the differences observed is 3.6 ± 3.2 x 10$^{14}$ molec cm$^{-2}$ or 11% ± 8%, on average, with a maximum of 2.0 x 10$^{15}$ molec cm$^{-2}$, again seen over the harbour of Antwerp."

**20) Page 19 Line 10: delete 'allow to'**

Corrected

**21) Figure 1: Adding a label for Stabroek as the other ground site where meteorology is measured in Antwerp could be helpful.**

Ok, a label was added for Stabroek, Antwerp.

**22) Figure 13: Please make the red dots more visible. (Perhaps white like in other Figures). Also in the caption write what they are. And as a suggestion, pull the color bar legend out of panel (a) and make larger since it refers to all four maps.**

We have made the red dots larger and white like in Fig. 6 and 7, and described it in the caption. We have extracted the legend from map a) and use it as a general legend for all maps. Note that we have put the different parts of the figure together in the word file. We will make a proper merged figure with the legend more central over the four plots for the final version.

For consistency we have applied the same to Figure 14 and its caption.

---

## Author Comment (AC2) · 7 Oct 2020

**Anonymous Referee #2:**

Thank you very much for the useful and constructive remarks. As described below, we have modified the manuscript according to suggestions and provided clarifications where necessary. We hope that the revised manuscript has improved in respect to the original paper. Please find a rebuttal against each point below.

*Black, bold, italic: Referee's comments*

Black: Author's reply

Changes in the original discussion paper are highlighted in yellow and attached below

***1) p.3 l. 31: Is this really the first? Around the same time: Evaluating Sentinel- 5P TROPOMI tropospheric NO2 column densities with airborne and Pandora spectrometers near New York City and Long Island Sound, Laura M. Judd et al.; https://doi.org/10.5194/amt-2020-151; I would suggest deleting this comment and possibly include a reference to this paper within the manuscript.***

The study https://doi.org/10.5194/amt-2020-**151** was indeed submitted to AMT in the same week as the study under review (https://doi.org/10.5194/amt-2020-**148**). We have adapted the paragraph in the manuscript and we have added a proper reference now it is available:

This is one of the first publications assessing TROPOMI $NO_2$ retrievals over strongly polluted regions based on the comparison with airborne remote sensing observations and it is one of the first airborne spectrometer data sets coinciding in space and time with a large amount of fully sampled satellite pixels. At the same time the study of Judd et al. (2020) on the Long Island Sound Tropospheric Ozone Study (LISTOS) campaign in the New York City/Long Island Sound region has been submitted. Earlier studies reporting on the validation of spaceborne observations based on airborne spectrometer data, such as Heue et al. (2005), Constantin et al. (2016), Lamsal et al. (2017), Broccardo et al. (2018), and Merlaud et al. (2020) have shown high potential but are scarce, mainly due to the relatively large pixel footprint of TROPOMI's predecessors with respect to the area that can be covered with an airborne mapping spectrometer.

However, we would find it more than fair that in the study https://doi.org/10.5194/amt-2020-151, the statement "This is the first airborne spectrometer dataset to be used to evaluate the TROPOMI tropospheric NO2 product." would be changed as well and a proper reference to this study would be added.

***2) p. 4 l. 16: The TROPOMI tropospheric columns are up to _12km. There is NO2 above 6.5km, the NO2 profile is not 0. Over cities and enhanced areas this will not be a big factor, but this should be discussed and mentioned. A typical amount of NO2 from 6.5 to 12km over Belgium would be useful to mention – maybe using CAMS or TMP.***

Thanks for the interesting comment. As suggested we checked the partial $NO_2$ column between 6.5 km and tropopause, based on the interpolated TM5-MP a priori $NO_2$ profiles over Brussels and Antwerp for the four campaign days. The TM5-MP a priori $NO_2$ profiles are provided in Fig. 3a (the figure was updated in the manuscript as there was a mistake in the plot in the conversion from VMR to partial columns). Note that the tropopause is around 16 km (defined from the temperature

profile) instead of the suggested 12 km on these days. The partial column between 6.5 and 16 km ranges between 2.8 and 4.7 x $10^{14}$ molec cm$^{-2}$. We can also refer to a similar question/answer (comment 2) in https://amt.copernicus.org/preprints/amt-2020-151/amt-2020-151-AC1-supplement.pdf. Judd et al. (2020) reports a partial column of 2 x $10^{14}$ molec cm$^{-2}$, but the aircraft altitude is 9 km instead of 6.5 km.

The impact on the comparison and conclusions is expected to be small and would generally increase the bias between the TROPOMI NO$_2$ VCD product and APEX retrievals, as the latter could be underestimated. Note that the effective impact is difficult to assess as airborne measurements are in fact sensitive to NO$_2$ above the flight altitude of 6.5 km, however, indeed with reduced sensitivity as can be observed in Fig. 3b. Retrieved SCDs are the sum of the measured differential slant column and the residual amount of NO$_2$ in a reference spectrum acquired over a clean area during the same flight (SCD = DSCD + SCDref). The residual amount in the reference spectrum is a tropospheric VCD (corrected for the stratospheric content) estimated in this work from mobile DOAS measurements (but can also be derived from for example a model or MAX-DOAS observations like done in a number of other studies). In principle SCDref contains implicitly a contribution from the upper troposphere (> 6.5 km). However, also these measurements, similar to MAX-DOAS measurements, have a reduced sensitivity to the upper troposphere. In case there are temporal/spatial changes in the NO$_2$ field in the upper troposphere between reference area and measured area this should be implicitly measured in the DSCD.

We have added a discussion in the manuscript in Sect. 5.2.2: "Note that APEX observations have reduced sensitivity to the NO$_2$ above the aircraft altitude of 6.5 km (see Fig. 3b), while the TROPOMI NO$_2$ VCD is defined up to the tropopause (approximately 16 km on the campaign days). The TM5-MP NO$_2$ partial columns between 6.5 and 16 km range between 2.8 and 4.7 x $10^{14}$ molec cm$^{-2}$. Retrieved APEX SCDs are the sum of the measured differential slant column and the residual amount of NO$_2$ in a reference spectrum acquired over a clean area during the same flight. SCDref is derived from a tropospheric VCD, estimated in this work from mobile DOAS measurements. In principle SCDref contains implicitly a contribution from the upper troposphere. However, also these measurements have a reduced sensitivity to the upper troposphere. In case there are temporal or spatial changes in the NO$_2$ field in the upper troposphere between the reference area and observed area, this should be implicitly measured in the DSCD. As the amount of NO$_2$ in the upper troposphere appears to be small compared to the total column over polluted sites and as the APEX retrievals still have some sensitivity to it, we expect any impact on the comparisons to be minimal."

*3) p. 4/5 I think the cloud fraction should be mentioned. It's mentioned that for the flights it was mainly clear sky, but what is the range of the cloud fractions for the TROPOMI observations? Some of this could be of course due to aerosols, but I think it would be good to know the cloud fraction (nitrogen dioxide window) assumed in the TROPOMI retrieval. I just noticed this is mentioned later in Sect. 4.3.3 , but it would be good to include it in this section.*

Ok, we have specified the TROPOMI cloud fraction in Section 2: "Flights took place in mostly cloud-free conditions and on days with good visibility. For flights on 27 to 29 June, there was a cloud fraction of less than 1% for the TROPOMI NO$_2$ retrieval window at 440 nm. Only on 26 June (Flight #1), conditions were not fully optimal with few scattered clouds and some light haze and aerosols (cloud fraction of 12%)."

*4) p.10 l. 1 Could the difference of the AMF come from the different height? APEX is from the surface to 6.5km; for TROPOMI it's higher.*

It has certainly an effect on the tropospheric AMF: due to scattering and absorption, the sensitivity to NO$_2$ decreases towards the ground surface and the decrease in sensitivity is stronger with increasing

platform altitude due to the larger scattering probability above the absorbing layer. See for example Fig. 8 in https://doi.org/10.5194/amt-12-211-2019.

We have added the following sentence to clarify this: "This can be partly explained by a stronger decrease in sensitivity with increasing platform altitude due to the larger scattering probability above the absorbing layer."

**5) Sect 4.3.2 Albedo: the TROPOMI AMF could be re-calculated using the APEX albedo and the impact can be directly estimated. I think the study would benefit from looking at the impact of correcting for the albedo. I'm not sure if this would be possible to do within a reasonable amount of time, if this is too time consuming, just mention it at least.**

Recalculating the TROPOMI tropospheric AMF (and $NO_2$ VCDs) based on the APEX albedo is not as straightforward as for example replacing the a priori $NO_2$ profiles based on the provided AKs, especially as the main authors of this study are not involved in the operational TROPOMI $NO_2$ retrievals. We agree it would be an added-value to directly study the impact on the $NO_2$ retrievals, instead of the albedo comparison tests done in this study. However, we propose to keep this as part of a future study: new APEX flights are foreseen over the two target areas in summer 2021 (and also other validation activities will take place later this year and next year → see reply to comment 8). As the TROPOMI LER (under development – see last paragraph on page 11) should be available by then, we suggest to compare the new APEX retrievals with the TROPOMI retrievals based on the initial OMI LER and new TROPOMI LER product to asses the impact.

To make clear that a study on the direct impact of the albedo is in the pipeline we have added the following at the end of the last paragraph on page 11: "New APEX validation flights over the Antwerp and Brussels region are foreseen for summer 2021 and will be valuable to assess 1) the retrieval impact of replacing the OMI LER by the TROPOMI LER, and 2) the v2 reprocessing of the TROPOMI $NO_2$ product".

**6) Sect. 4.3.2: albedo is wavelength dependent; albedos at 3 different wavelengths are compared The. How big is the impact of? the wavelengths difference This should be discussed, e.g. look at the OMI albedo and include the relative difference for these different wavelengths (over Brussels and Antwerp).**

The albedo is indeed wavelength dependent and therefore a statement was present in the next to last paragraph to warn for the difficulties when comparing different albedo products: "Even if a direct comparison of different albedo products is not trivial due to BRDF-effects and albedo wavelength dependencies, among other…"

We followed your suggestion and tried to quantify the wavelength dependency based on the OMI LER. Below is a plot of the surface reflectance over Antwerp and Brussels for the 23 wavelength bands (and for both the yearly and monthly OMI LER product (June)). For TROPOMI $NO_2$ retrievals, the OMI LER product at 440 nm is used. Note that we used the MODIS MCD43A3 product at 470 nm and APEX SR is at 490 nm (middle of the APEX $NO_2$ fitting interval). The relative differences are 0 for the pixel over Brussels and 2.3% (yearly) – 3.8% (monthly) over Antwerp. Note that the relative difference between 440-470 and 440-490 is the same in all cases. Based on these tests, the impact of the wavelength dependency seems to be small at these wavelengths.

[Figure]

We have added the following in the paper: "The albedo is wavelength dependent and albedo products at different wavelengths have been compared: OMI LER at 440 nm, MODIS MCD43A3 at 470 nm and APEX albedo at 490 nm. The wavelength dependency has been assessed by analysing the relative difference of the OMI LER albedo over Brussels and Antwerp between 440 nm, and 470-490 nm for both the yearly and monthly OMI LER product (June). Overall, the OMI LER albedo increases slightly with wavelength but the increase is smaller than 4% between 440 and 490 nm."

***7) p. 13, l. 16: it could be due to meteorology; e.g. lower wind speeds can increase the VCD enhancement even though emissions do not increase VCDs can be higher for stagnant winds, there could also be factors that potentially increased the lifetime of NOx (e.g. OH, O3, and NOx concentration) for that particular day. I think meteorology should be mentioned as a potential influence; look at the wind speeds and direction for these days (the wind speed is definitely lower). If both TROPOMI and APEX observed higher VCDs on June 29, this would not be due to the APEX instrument troubles.***

Good comment! Indeed, meteorology is certainly playing a role here, as well as other factors increasing the NOx lifetime. I was surprised at first glance not to see a "weekend effect", which we usually see for example in MAX-DOAS data in Brussels, mainly monitoring traffic emissions. However, thinking about it we are mainly looking at emissions from petrochemical industry, which is probably constant each day of the week…this in contrast to traffic emissions. And like you comment lower wind speeds and other factors can increase the VCD, even when emissions are stable. We have altered the paragraph as follows:" Although June 29 is a Saturday, the $NO_2$ VCDs observed over the Antwerp harbour are slightly higher than on June 27, both in the APEX and TROPOMI data. The prevailing emissions in Antwerp from petrochemical industry are expected to be rather constant in contrast to traffic emissions, but meteorology, for example a more stagnant wind speed (3.7 m s$^{-1}$ on 27 June and 2.3 m s$^{-1}$ on 29 June, on average), and other factors that can potentially increase the lifetime of NOx, might explain the slight $NO_2$ VCD increase observed on June 29. However, when …

Note that there is some misunderstanding here: the instabilities with the instrument are no explanation for the fact that we observe higher VCDs on June 29. We mentioned this to explain why we don't have data over the city center on June 29, as we couldn't analyse the first three flight lines. The data we show and compared with TROPOMI are not affected by the encountered instrumental

issues. In fact both were 2 different remarks related to the 29 June VCD map and we suggest to split it in two (small) paragraphs for clarity.

*8) p. 22, l. 21: Can you really conclude this if your comparison is done over a small area and over a short time period, seasons are not considered (e.g. snow)? Re-phrase this, or add "over Belgium in the summer time."*

Thanks for pointing this out. We agree that the statement is "too strong" based on the data set we currently have. We have nuanced this in the conclusion: "The case study over polluted regions in Belgium in summer time demonstrates that the TROPOMI tropospheric $NO_2$ product meets the mission requirements in terms of precision and accuracy."

Note however that some nuance was already present in the last paragraph of the conclusion, i.e. mentioning that more flights/data are needed under different geophysical conditions: "…The main focus was to quantify the TROPOMI retrieval uncertainties in polluted regions and results from the comparison with APEX data, acquired over Belgium in summer time, have shown that the TROPOMI tropospheric $NO_2$ product meets the mission requirements in terms of accuracy and precision. However, additional validation studies are required and are currently planned, focusing on more sites with different geophysical properties and varying pollution levels, including background areas, extreme albedo sites, other seasons, and cloudy scenes, among others, in order to assess as well the performance in suchlike conditions."

Note as well that new flights will take place over the two sites in summer 2021. Also other validation activities, involving airborne instruments, will take place later this year and next year over several sites in Europe. For example, recurrent flights will take place over the cities of Bucharest and Berlin covering different seasons. Still under construction but here's a link to upcoming TROPOMI validation activities: https://s5pcampaigns.aeronomie.be/

*9) p.1 l. 24-26: "When the absolute value . . ., when comparing APEX NO2 VCDs with TM5-MP based and CAMS-based NO2 VCDs, respectively." I suggest re-wording this sentence, e.g.: The absolute difference is on average xx molec cm-2 (16%) and xx molec cm-2 (9%) compared to . . .*

We have applied the suggestion in the abstract (and also in the conclusion).

*10) p.1 l. 26: Which accuracy requirement; maybe change it to "mission accuracy requirement"*

Good suggestion and corrected throughout the paper.

*11) p.1 l. 29-30; suggest re-wording: Something like: The current TROPOMI data underestimate localized enhancements and overestimate background values by approximately 1-2x 1015 molec cm-2 (10- 20%).*

Thank you for the suggestion! Note that for the same reason as for the earlier comment #8 (and also comment #8 from Reviewer #1) we have also added some nuance.
In the abstract: "For a case study in the Antwerp region, the current TROPOMI data underestimates localised enhancements and overestimates background values by approximately $1\text{-}2 \times 10^{15}$ molec $cm^{-2}$ (10- 20%)."

In the conclusion:" The TROPOMI spatial resolution is limited to resolve fine-scale urban $NO_2$ plumes and can cause a considerable smoothing effect in case of the observation of strongly polluted scenes

with steep gradients. This depends both on the instrument pixel size and the amount of heterogeneity in the $NO_2$ field. The high-resolution APEX retrievals allow to monitor the effective horizontal variability in the $NO_2$ field at much finer scale. In Sect. 6, the impact of smearing of the effective signal due to the finite satellite pixel size was studied for the Antwerp region based on a downsampling approach of the APEX retrievals. Assuming a pixel size of 25 to 20 $km^2$, equivalent to the initial 3.5 km x 7 km and new TROPOMI 3.5 km x 5.5 km spatial resolution (at nadir), the TROPOMI data underestimates localised enhancements and overestimates urban background values by approximately 1-2 x $10^{15}$ molec $cm^{-2}$, on average, or 10% - 20%, for the Antwerp case study. The average under- and overestimation is further reduced to 0.6-0.9 x $10^{15}$ molec $cm^{-2}$, or smaller than 10%, when increasing the pixel size to 1 $km^2$. Therefore, detailed air quality studies at the city scale still require observations at higher spatial resolution, in the order of 1 $km^2$ or better, in order to resolve all fine-scale structures within the typical heterogeneous $NO_2$ field."

*12) p.3 l. 13: "studied in Sect. 6" change to "see Sect. 6"*

Corrected.

*13) p.4 l. 4 Air pollution levels over Belgium. . . Do you have a reference that can be included here?*

This statement was mainly based on own experience while looking at satellite data, where in Europe always a hotspot over North of Belgium/South of The Netherlands is popping up, together wit the Po valley, Paris, Ruhr area, etc.

I couldn't find a proper peer reviewed journal paper but we suggest to add following reference from a Greenpeace study where Flanders, or more specifically Antwerp, is mentioned as one of the 50 biggest $NO_2$ hotspots in te world. Only the German Ruhr and Paris are the other two European hotspots appearing in the list:

https://www.greenpeace.org.au/research/new-satellite-data-reveals-worlds-largest-air-pollution-emission-hotspots-greenpeace-media-briefing/

also the in-situ data from the environmental network on following website is showing how Belgium is flirting with or exceeding the EU annual limit value of 40 $\mu$g/m3 for $NO_2$:

https://www.eea.europa.eu/themes/air/country-fact-sheets/2019-country-fact-sheets/belgium-air-pollution-country

*14) p.6, l. 16 VNIR; this should be defined, maybe in the previous sentence were the two channels are mentioned*

As suggested, we have changed the previous sentence to: "APEX records backscattered solar radiation in the visible, (near-)infrared regions of the electromagnetic spectrum, covering the 370 to 2540 nm wavelength range in two channels, a visible/near-infrared channel (VNIR) and a short-wave infrared channel (SWIR)."

*15) p.7 l. 28, mention the height of the layers (between surface and xx km)*

Profile height has been added: "...and (2) daily $NO_2$ vertical profiles from the TM5-MP model on a 1° × 1° grid and covering 34 vertical layers (between surface and TOA)."

---

## Author Response (AR2)

**Dear Editor, dear Anonymous Referee #1:**

We greatly appreciate the constructive comments. As described below, we have modified the manuscript according to suggestions and clarified where necessary. We hope that the revised manuscript has improved in respect to the original paper. Please find a rebuttal against each point below.

*Black, bold, italic: Referee's comments*

Black: Author's reply

Changes in the original discussion paper are highlighted in yellow and attached below

*1) On pages 3-4, line starting at 33 the statement "At the same time the study of Judd et al. (2020) on the Long Island Sound Tropospheric Ozone Study (LISTOS) campaign in the New York City/Long Island Sound region has been submitted". I find the statement little odd in a publication. Please note that the manuscript by Judd et al is already published in AMT. Maybe you could have a statement like "A similar study was carried out by Judd et al. (2020) using observations from the Long Island Sound Tropospheric Ozone Study (LISTOS) campaign in the New York City/Long Island Sound region in the United States". In my opinion, both of the studies (Tack et al and Judd et al) are very helpful in accessing the quality of TROPOMI NO2 retrievals and unraveling complexities associated with validation of satellite NO2 retrievals in different environments and possibly different seasons. I encourage you to cite Judd et al and make those connections as appropriate.*

Thank you! We have followed the suggestion. We have updated the reference to the published paper in AMT. Note that we also updated the conclusion and included a comparison of our findings with the findings in the study of Judd et al. (2020):

"Nevertheless, TROPOMI is generally biased low over polluted areas when compared to ground-based or airborne observations and this is consistent with the findings in other studies, such as Griffin et al. (2019), Ialongo et al. (2019), Zhao et al. (2019), Dimitropoulou et al. (2020), Judd et al. (2020), and Verhoelst et al. (2020). The study of Judd et al. (2020) is based on 16 flight days with GeoTASO and GCAS over the New York City/Long Island Sound region. Overall, the standard TROPOMI NO$_2$ VCD product is very well correlated (R = 0.96), but negatively biased by approximately -19% with respect to coincident GeoTASO/GCAS NO$_2$ retrievals. Replacing TM5-MP a priori NO$_2$ profiles by NO$_2$ profile shapes from the North American Model–Community Multiscale Air Quality model (NAM-CMAQ) at 12 km spatial resolution improves the overall bias to -7%. ..."

*2) On page 12, line 13: I think the reference "Henk Eskes, personal communication, 15 March, 2020)" is unnecessary as Henk Eskes is also a co-author. Please check.*

In this context, we had cited Henk rather as the PI of the TROPOMI tropospheric NO2 product than as a co-author of the paper. Following the comment we have removed the source for this statement.
* * *
*1) Line 5 page 2: Technically the temporal frequency of TROPOMI is not unprecedented, so edit 'spatio-temporal' to just 'spatial'. It is true that the quality of data allows easier use of daily data than its predecessors that require averaging of many overpasses, but this isn't stated and as written it could be confusing to a reader thinking that TROPOMI measures more frequently.*

Corrected

*2) Page 17 Line 22: The airborne data picks up fine spatial structure, but when sampled to the TROPOMI pixel, this should be taken care of through the spatial averaging and not a source of low bias, correct? Therefore, this statement should be taken out here. It would be more appropriate to state something like this when discussing the right side of the figure where the impact of undersampling is clearly shown as an impact.*

Yes, we understand your point of view. Technically it could play a role as the statistics are based on TROPOMI pixels that are covered at least by 50% by APEX observations. But as stated later on in the paragraph the statistics do not change when we consider only fully covered TROPOMI pixels. Therefore we have followed the above remark in the revised version. Note that this is also corrected in the conclusion.

*3) Page 20 Line 17: Technically Pandora can operate in MAX-DOAS mode so maybe clarify as Pandora direct-sun measurements.*

Corrected

*4) There is a definition late in the paper about what urban background is: 'Note that in this work urban background is defined as an area in a polluted environment, which is not directly affected by pollution plumes'. However, it is not clear how data is 'classified' as urban background. Can this be clarified? Especially with the discussion in Sect 6.*

By visual inspection of the $NO_2$ maps it can be easily identified if an area is affected by an urban / industrial plume or not and if it has low gradients in the $NO_2$ field.

*5) Page 4 Line 10 and Figure 1: I see 6 red markers rather than five. One is in France so perhaps this can be mentioned in the caption and text.*

Corrected in the text and the caption.

*6) Figure 1: The arrows are 180 deg off from the wind direction (table says the wind is from 36 deg), which I later put together that they were just pointing out emission sources and not the wind vector coming from the emission source. Considering flipping these arrows or make it more obvious that it is not a wind direction.*

Corrected

**7) Line 19 page 2: Please clarify what it means by having potential. Potential for what?**

It is referring to the previous paragraph where campaign concepts were identified and prioritized based on a set of validation requirements. We have reformulated this to:

“On this basis, a S-5P validation campaign over Belgium (S5PVAL-BE), focusing on nitrogen dioxide ($NO_2$) column airborne observations, was identified as having much potential and high priority for TROPOMI validation due to…”

**8) Page 6 line 18: Should (near)infrared be (shortwave)infrared?**

Indeed! Corrected

**9) Page 3 line 30: It says that a TROPOMI pixel comprises of 4000 APEX pixels but throughout the rest of the paper it states 2700. Is this an error or is there a caveat to this that should be clarified when these numbers are stated.**

A **nadir** TROPOMI (3.5 km by 7 km) pixel corresponds to approximately 2700 APEX pixels, but the TROPOMI pixel size is changing (increasing) along the swath. Depending on the overpass, TROPOMI pixels are covered by 2700 to 4000 APEX pixels. We have clarified this in the abstract and conclusion. If we mention 2700 APEX pixels we specifically refer to a 3.5 km by 7 km TROPOMI pixel.

On p.3 L.30, this was indeed wrongly stated as we specifically refer to a 3.5 km by 7 km TROPOMI pixels. We have corrected this to 2700 pixels.

**10) Page 10 line 8: It is unclear what the 50% larger than is referring to. Is it that APEX AMF is 50% larger than TROPOMI with this box profile?**

The APEX box AMF is 50% larger than the TROPOMI box AMF for the layer at the ground surface. This layer has a thickness of 50 m. Note that this is referring to the box AMFs or vertical sensitivity to $NO_2$ and not to the vertical $NO_2$ profile.

We have clarified this in the revised version of the manuscript: “For a low albedo case, i.e. 0.02, the box AMF of the layer closest to the ground surface (50 m thickness) is ~50% larger for APEX when compared to TROPOMI, while this is ~15% for a very high albedo case, i.e. 0.2.”

**11) Page 10 line 27: The phrase 'When considering the entire APEX scenes instead of single pixels,' then going on about APEX pixels resampled to TROPOMI, I am confused what entire apex scene means for this sentence? I expect other readers would be too. Would the point still be clear if the phrase was removed?**

We agree this is badly formulated and is confusing. We hope that the revised phrases are more clear:

“Analysing the ensemble of the four acquired APEX data sets provided in Table 1 at the native APEX spatial resolution of 75 m x 120 m, the APEX albedo is 0.040 on average and the variability within one

TROPOMI pixel, expressed as the standard deviation (SD), is 0.022 on average or ~55%, but can be up to 100% for certain pixels. When resampled to the spatial resolution of TROPOMI, the variability of the APEX-derived albedo is 0.012 on average or ~30%, with values ranging between 0.015 and 0.065 (see Fig. 4)… "

*12) Page 16 line 12: remove 'the latter'*

Corrected

*13) Bottom of page 19: Does this mean that the LOS sampling of aircraft data is 10km too? Can the authors state explicitly how far out they integrate data out in the LOS?*

[revised manuscript text omitted]